# Deep Ensemble Clustering for Visual Representation Learning

**Yuwei Wang** [* 1]   **Guikun Chen** [* 2]   **Xiruo Jiang** [3]   **Yazhou Yao** [1]
**Di Liu** [4]   **Xiangbo Shu** [1]   **Fumin Shen** [5]   **Wenguan Wang** [2]

https://github.com/NUST-Machine-Intelligence-Laboratory/EnFormer

## Abstract

Recent advances in visual representation learning have seen the rise of clustering-based vision backbones, which adopt clustering as a core paradigm for feature extraction. However, existing clustering-based backbones typically rely on a single clustering algorithm, whose inherent inductive bias limits their representational capacity. To address this, we propose ENFORMER, which embeds ensemble clustering as a core component of feature extraction. ENFORMER structures feature extraction around two steps: (**i**) Ensemble Generation, where several differentiable base clustering methods are introduced to capture diverse semantic structures; and (**ii**) Consensus Aggregation, which employs a differentiable mechanism to fuse the results of all base clusterings to reconstruct refined visual features. Extensive experiments show that ENFORMER consistently outperforms existing clustering-based backbones across core vision tasks, with higher performance and significantly improved throughput.

## 1. Introduction

Visual representation learning, which aims to extract meaningful and abstract features from raw pixel data, stands as a cornerstone of modern computer vision. For years, this field has been driven by two dominant architectural paradigms: Convolutional Neural Networks (CNNs) (Krizhevsky et al., 2012; He et al., 2016) and Vision Transformers (ViTs) (Dosovitskiy et al., 2020; Liu et al., 2021b). More recently, a promising new direction has taken shape by framing representation learning through cluster-

---

[*]Equal contribution [1]Nanjing University of Science and Technology [2]The State Key Lab of Brain-Machine Intelligence, Zhejiang University [3]Southwest Jiaotong University [4]Southeast University [5]University of Electronic Science and Technology of China. Correspondence to: Yazhou Yao <yazhou.yao@njust.edu.cn>.

*Proceedings of the $43^{rd}$ International Conference on Machine Learning*, Seoul, South Korea. PMLR 306, 2026. Copyright 2026 by the author(s).

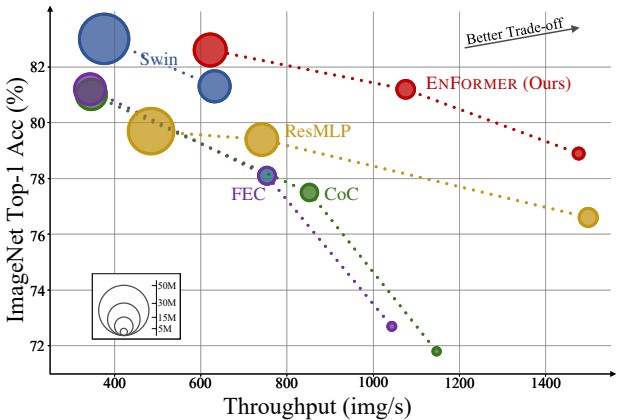

*Figure 1.* Top-1 accuracy *vs.* throughput on ImageNet-1K (Deng et al., 2009) `val` set. The bubble size corresponds to parameters. ENFORMER achieves a superior performance-efficiency trade-off compared to existing clustering-based methods (*i.e.*, CoC (Ma et al., 2023) and FEC (Chen et al., 2024)) and other methods (*e.g.*, Swin (Liu et al., 2021b) and ResMLP (Touvron et al., 2022)).

ing (Ma et al., 2023). This nascent paradigm conceptualizes feature extraction as a human-understandable process of grouping similar features into coherent clusters (Chen et al., 2024). Such an approach is particularly appealing for its inherent transparency, as the model's internal organization of the visual world (Lu et al., 2021) can be more readily inspected and understood (Wang et al., 2023b).

While these pioneering clustering-based backbones lay a solid foundation, their representational capabilities are shaped by the properties of one single clustering algorithm they employ. It is a well-established principle that *each single clustering algorithm carries an intrinsic inductive bias* (Mitchell, 1980), stemming from its specific optimization objective and assumptions about data structure (Fred & Jain, 2005; Domeniconi & Al-Razgan, 2009). The "No Free Lunch" theorem (Wolpert, 1996) further formalizes this trade-off: a model's necessary bias enables it to converge on a solution, yet restricts its expressive capacity to a specific subset of problems. To address this inherent limitation of single clustering, classical machine learning embraces ensemble clustering as a robust and powerful solution (Strehl & Ghosh, 2002; Fred & Jain, 2005). Concretely, ensemble clustering based approaches aggregate a diverse

collection of base clusterings to mitigate the systematic errors of individual clustering algorithms (Dietterich, 2000) and produces a consensus partition to improve quality and stability (Dudoit & Fridlyand, 2003).

Inspired by the principle of classical ensemble clustering, we propose ENFORMER, a new vision backbone architecture that embeds ensemble clustering directly into the feature encoding process. This design allows the model to construct visual representation by synthesizing multiple structural perspectives, rather than being confined to the specific perspective of one single algorithm. ENFORMER structures visual representation learning around two core steps: ensemble generation and consensus aggregation. During *ensemble generation*, several differentiable base clustering methods (*e.g.*, partitional, fuzzy, possibilistic, and probabilistic clustering) are introduced to produce cluster representations of different structures and capture distinct semantic relationships among image regions. This step is achieved by projecting features into multiple subspaces, each equipped with a dedicated clustering mechanism. In the *consensus aggregation* step, all cluster representations are concatenated into a unified dictionary of learnable structures. A differentiable consensus mechanism is proposed as an alternative to the traditional co-association matrix (Fred & Jain, 2005), adaptively weighting cluster representations and guiding the redistribution across the visual features. As a result, the entire process forms an end-to-end differentiable ensemble clustering based backbone for visual feature extraction.

The main characteristics of ENFORMER are four-fold. *First*, ENFORMER can integrate multiple heterogeneous clustering mechanisms to capture distinct and complementary *structural perspectives*, yielding richer and more robust feature representations. *Second*, despite its ensemble nature, EN-FORMER maintains computational *efficiency* comparable to a single-clustering backbone through balanced feature allocation and lightweight design. *Third*, both the base clustering and consensus components are *differentiable*, enabling end-to-end joint optimization within deep learning frameworks while implicitly adapting the contribution of each clustering through the learned consensus weighting. *Finally*, the proposed ensembling architecture is *modular* and easily extensible to allow new clustering algorithms to be incorporated without redesigning the overall framework.

To the best of our knowledge, ENFORMER is the first architecture to embed and integrate multiple clustering algorithms as core architectural primitives of backbones. We evaluate ENFORMER on three core computer vision benchmarks (*i.e.*, ImageNet-1K (Deng et al., 2009), MS COCO (Lin et al., 2014), ADE20K (Zhou et al., 2017)) to validate the effectiveness, where it consistently outperforms existing clustering-based counterparts. For example, our largest model, ENFORMER-Large, improves upon these

counterparts by **+1.4%∼1.6%** Top-1 accuracy on ImageNet-1K, **+3.4∼4.1** box mAP and **+2.4∼2.7** mask mAP on MS COCO, and **+5.8∼6.1** mIoU on ADE20K over these counterparts, while delivering up to **1.8×** higher throughput.

## 2. Related Work

**Vision Backbone Architectures.** Vision backbones serve as the fundamental architectures for feature extraction, with the current landscape of paradigms dominated by CNNs and ViTs. Following the success of AlexNet (Krizhevsky et al., 2012), landmark CNNs like InceptionNet (Szegedy et al., 2015) and ResNet (He et al., 2016) establish foundational design principles. This spurs extensive innovation focused on efficiency (Howard et al., 2017; Tang et al., 2022; Chen et al., 2023; Yang et al., 2026), receptive field design (Liu et al., 2022; Ding et al., 2022), and flexible convolution operators (Dai et al., 2017; Wang et al., 2023a; Ding et al., 2021). ViT (Dosovitskiy et al., 2020) introduces a new paradigm based on global self-attention. The initial limitations of ViT, such as quadratic complexity and a non-hierarchical structure, prompt refinements like hierarchical designs (Wang et al., 2021; Liu et al., 2021b), local-global strategies (Chu et al., 2021; Yang et al., 2021), and hybrid CNN-ViT architectures (Rao et al., 2021; d'Ascoli et al., 2021; Xu et al., 2021). Meanwhile, alternative architectures including MLP-based models (Tolstikhin et al., 2021; Yu et al., 2022; Touvron et al., 2022) and clustering-based models (Ma et al., 2023; Liang et al., 2023; Chen et al., 2024) have also emerged as competitive paradigms. The clustering-based models frame representation learning as a human-understandable process of feature aggregation and assignment, providing insight into how representations are formed within backbones. Building upon this clustering-based paradigm, ENFORMER preserves its inherent transparence while enriching the capacity to model diverse structural relations among visual features.

**Deep Clustering.** Clustering, a cornerstone of machine learning, provides a powerful technique for uncovering latent structures in data without supervision. The principles of clustering have been integrated into the deep learning framework, thereby extending its utility and driving applications across diverse domains, including medical and biological science (Tian et al., 2019; Quan et al., 2024), social media analysis (Bianchi et al., 2020; Yadav & Vishwakarma, 2020), and 3D perception (Yin et al., 2022; Feng et al., 2023; 2024). In computer vision, clustering is central to a wide spectrum of works, *e.g.*, image segmentation (Shi & Malik, 2000; Ren & Malik, 2003; Yin et al., 2026a;b), generative modeling (van den Oord & Vinyals, 2017; Mukherjee et al., 2019), and unsupervised representation learning (Caron et al., 2018; 2020; Asano et al., 2020; Caron et al., 2021). More recently, researchers begin to embed clustering directly into vision

backbones, utilizing it as a mechanism to extract feature information for supervised representation learning (Ma et al., 2023; Liang et al., 2023; Chen et al., 2024). This evolution marks the elevation of clustering above and beyond a mere analytical tool to a core component for modeling complex, high-dimensional data relationships.

While these methods have established the value of clustering in deep learning, their representational capabilities are constrained by using a single clustering algorithm. This limitation might compel the model to learn suboptimal features. Therefore, ENFORMER harnesses the complementary strengths of multiple clustering algorithms to mitigate suboptimality. This represents a shift from the single clustering paradigm, leading to more robust and stable representations.

**Ensemble Clustering** aims to produce a consensus partition that is more robust and accurate than any individual solution by combining multiple base clusterings. The traditional two-stage process involves ensemble generation and consensus aggregation. Ensemble generation introduces diversity via algorithmic variations (Dudoit & Fridlyand, 2003; Fred & Jain, 2005; Hadjitodorov et al., 2006; Kuncheva & Vetrov, 2006), partitioning various feature subspaces (Strehl & Ghosh, 2002) or operating on different subsets of the data instances (Domeniconi & Al-Razgan, 2009; Dudoit & Fridlyand, 2003). Consensus aggregation then merges these clustering partitions using strategies like graph-based learning (Huang et al., 2017; Zhou et al., 2021a;b), matrix factorization (Li et al., 2007), or enhanced co-association matrices (Huang et al., 2018; Jia et al., 2024).

ENFORMER takes the first step to embed ensemble clustering as a core block for end-to-end, supervised visual representation learning. We introduce a differentiable consensus mechanism that aligns clustering objectives with feature learning and enables joint optimization within a unified deep framework. Furthermore, this mechanism adaptively weights clusters to suppress unreliable ones, consequently improving the overall performance. Our work establishes ensemble clustering as a powerful and native component within modern vision architectures.

## 3. Methodology

Our goal is to develop a deep ensemble clustering framework that learns visual representations by integrating multiple clustering methods to mitigate the inherent bias of individual clustering algorithms. To achieve this, we propose ENFORMER, a vision backbone composed of several Ensemble Blocks integrating the core Deep Ensemble Clustering (DEC) module, as shown in Fig. 2.

In the following, we first present the conceptual foundation behind ENFORMER (§3.1), then detail the design of DEC (§3.2), and finally provide analytical discussions (§3.3).

### 3.1. Clustering Perspective on Visual Representation Learning

Whereas CNNs and ViTs process images as structured grids of pixels or sequences of patches, a promising direction in visual representation learning is to conceptualize *images as a set of discriminative points*. To enable parallel computation, these points are represented as a feature matrix $\boldsymbol{P} \in \mathbb{R}^{N \times D}$. The term "discriminative" here implies that the points possess latent properties that allow them to form meaningful groups. This characteristic is used to formulate feature encoding as a process of clustering based information exchange. Concretely, rather than serving as a final output, these dynamically formed groups act as conduits for redistributing contextual information among points. Such an encoding mechanism transforms $\boldsymbol{P}$ into an enhanced representation $\boldsymbol{P}' \in \mathbb{R}^{N \times D}$, which is organized as:

$$
\begin{aligned}
\text{Initialization: } & \boldsymbol{O} \leftarrow \texttt{AdaPool}_M(\boldsymbol{P}), \\
\text{Assignment: } & \boldsymbol{A} \leftarrow \texttt{Norm}(\texttt{Sim}(\boldsymbol{O}, \boldsymbol{P})), \\
\text{Aggregation: } & \boldsymbol{C} \leftarrow \boldsymbol{A}\boldsymbol{P}, \\
\text{Redistribution: } & \boldsymbol{P}' \leftarrow \boldsymbol{P} + \boldsymbol{A}^{\top}\boldsymbol{C}.
\end{aligned}
\tag{1}
$$

This pipeline begins by initializing $M$ cluster centers $\boldsymbol{O} \in \mathbb{R}^{M \times D}$ from $\boldsymbol{P}$ via an adaptive pooling (`AdaPool`) operation. These centers are then used to generate an assignment matrix $\boldsymbol{A} \in [0, 1]^{M \times N}$, which quantifies the membership of each feature point in each cluster. Guided by $\boldsymbol{A}$, the content-aware cluster representations $\boldsymbol{C} \in \mathbb{R}^{M \times D}$ are aggregated from $\boldsymbol{P}$, which summarizes the features within each cluster. Finally, the summarized cluster information in $\boldsymbol{C}$ is redistributed back to individual feature points via a residual update to yield the enhanced representation $\boldsymbol{P}'$.

Building upon the above single clustering pipeline, we generalize the formulation to use an ensemble of several base clusterings to incorporate multiple structural perspectives. This ensemble process (§3.2) generalizes the Assignment and Aggregation phases (Eq.1) into a multi-clustering form, and is implemented in two differentiable steps: **i**) *Ensemble Generation*, which applies multiple, distinct clustering mechanisms to produce a diverse set of representations; and **ii**) *Consensus Aggregation*, which integrates these perspectives to reconstruct the final, unified features.

### 3.2. Deep Ensemble Clustering (DEC)

The DEC module, as the core of ENFORMER, is equipped with several differentiable base clustering methods and a differentiable consensus aggregation mechanism to mitigate the limitation of single clustering algorithm. This section first introduces the common initialization and similarity computation, then details each base clustering method, and finally describes how these components are combined through ensemble generation and consensus aggregation.

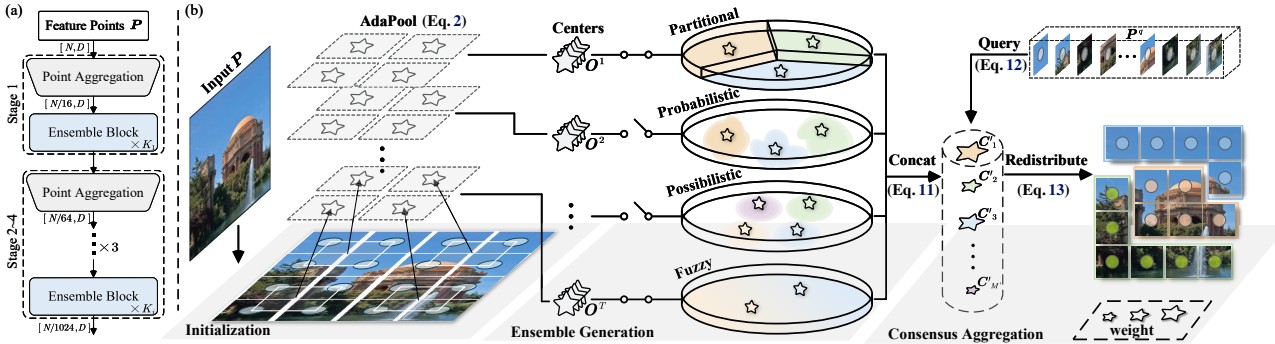

*Figure 2.* (a) Overall framework of ENFORMER. Each stage $i$ contains $K_i$ Ensemble Blocks. (b) Illustration of the Deep Ensemble Clustering module. The pipeline includes three steps: **Initialization** of cluster centers, **Ensemble Generation** applying several base clustering methods, and **Consensus Aggregation** integrating all cluster representations to reconstruct the unified features.

**Center Initialization and Similarity Computation.** Given feature points $\boldsymbol{P} \in \mathbb{R}^{N \times D}$, we project them into *key* and *value* spaces via linear transformations to obtain $\boldsymbol{P}^k, \boldsymbol{P}^v \in \mathbb{R}^{N \times D}$. For simplicity, we reuse $D$ to denote the projected channel dimension, as subsequent computations are performed entirely in these feature spaces. The cluster center matrices $\boldsymbol{O}^k, \boldsymbol{O}^v \in \mathbb{R}^{M \times D}$ are initialized by applying AdaPool over $M$ grid cells:

$$\begin{aligned}
\boldsymbol{O}^k &= [\boldsymbol{o}_1^k; \ldots; \boldsymbol{o}_M^k] = \texttt{AdaPool}_M(\boldsymbol{P}^k), \\
\boldsymbol{O}^v &= [\boldsymbol{o}_1^v; \ldots; \boldsymbol{o}_M^v] = \texttt{AdaPool}_M(\boldsymbol{P}^v),
\end{aligned} \quad (2)$$

where $\boldsymbol{o}_m^k / \boldsymbol{o}_m^v \in \mathbb{R}^D$ denotes the $m$-th cluster center vector in *key/value* space. We then measure the cosine similarity between the initialized centers and the feature points to obtain the similarity matrix $\boldsymbol{S} \in \mathbb{R}^{M \times N}$:

$$\boldsymbol{S} = \boldsymbol{\alpha} \odot \langle \boldsymbol{O}^k, \boldsymbol{P}^{k\top} \rangle + \boldsymbol{\beta}, \quad (3)$$

where $\odot$ denotes element-wise multiplication, $\langle \cdot, \cdot \rangle$ represents the cosine similarity, and $\boldsymbol{\alpha} \in \mathbb{R}^M$ and $\boldsymbol{\beta} \in \mathbb{R}^M$ are learnable parameters. Here, $\boldsymbol{\alpha}$ and $\boldsymbol{\beta}$ are broadcast to match the dimensions of the similarity computation. This affine calibration allows each cluster to adaptively rescale and shift its similarity scores, which improves optimization stability.

**Partitional Clustering** aims to partition features into distinct, non-overlapping groups (MacQueen, 1967). This base clustering method provides a definitive structural perspective by enforcing a hard assignment for each feature, making it particularly effective at identifying well-defined, separable structures. To enable gradient-based learning while retaining discrete assignments, we employ a straight-through Gumbel-Softmax estimator (Jang et al., 2016). Given the similarity matrix $\boldsymbol{S}$ (Eq. 3), the estimator produces a relaxed sample $\tilde{\boldsymbol{X}} = \text{GumbelSoftmax}(\boldsymbol{S}) \in [0,1]^{M \times N}$ to obtain its discretized counterpart $\boldsymbol{X} \in \{0,1\}^{M \times N}$. In addition, we use a sigmoid transformation to adjust assignment confidence, producing the final assignment matrix $\boldsymbol{A} \in [0,1)^{M \times N}$:

$$\boldsymbol{A} = \left( 1/(1 + \exp(-\boldsymbol{S})) \right) \odot \boldsymbol{X}. \quad (4)$$

This design combines the discrete cluster selection from $\boldsymbol{X}$ with continuous confidence weighting from $\boldsymbol{S}$, allowing differentiable yet structurally clear partitioning. The aggregated cluster representations $\boldsymbol{C} \in \mathbb{R}^{M \times D}$ are computed as:

$$\boldsymbol{C}_{ij} = (\boldsymbol{O}_{ij}^v + (\boldsymbol{A}\boldsymbol{P}^v)_{ij})/(\sum\nolimits_{n=1}^{N} \boldsymbol{X}_{in} + 1), \quad (5)$$

where the denominator provides normalization to avoid magnitude explosion and mitigate empty-cluster effects. The additive constant also acts as a safe guard when few points are selected, preventing unstable updates in early training.

**Fuzzy Clustering.** Drawing inspiration from Fuzzy C-Means (Bezdek et al., 1984), we design a differentiable fuzzy clustering to permit each point to belong to multiple clusters. This base clustering method provides a soft membership view that is capable of capturing overlapping or ambiguous structures. We apply a softmax function over the similarity matrix $\boldsymbol{S} \in \mathbb{R}^{M \times N}$ to compute the fuzzy assignment matrix $\boldsymbol{A} \in (0,1)^{M \times N}$, ensuring that each point's membership degrees sum to one across all clusters. Specifically, the fuzzy assignment matrix $\boldsymbol{A}$ is computed as:

$$\boldsymbol{A}_{ij} = \exp(\boldsymbol{S}_{ij}) / \sum\nolimits_{m=1}^{M} \exp(\boldsymbol{S}_{mj}). \quad (6)$$

Similar to Eq. 5, the cluster representations $\boldsymbol{C} \in \mathbb{R}^{M \times D}$ are computed by a weighted aggregation over all points:

$$\boldsymbol{C}_{ij} = (\boldsymbol{O}_{ij}^v + (\boldsymbol{A}\boldsymbol{P}^v)_{ij})/(\sum\nolimits_{n=1}^{N} \boldsymbol{A}_{in} + \epsilon), \quad (7)$$

where $\epsilon$ is a small constant to prevent division by zero. It is worth noting that the normalization strategy here differs from that in partitional clustering. Partitional clustering adopts hard assignments, where each point is assigned to a single cluster, and the denominator in Eq. 5 corresponds to the number of selected points according to the binary assignment matrix $\boldsymbol{X}$. In contrast, fuzzy clustering adopts soft assignments, where each point can contribute to multiple clusters with different membership weights. Therefore, the aggregation in Eq. 7 forms membership-weighted cluster

representatives, and the denominator is given by the total soft membership of each cluster in $\boldsymbol{A}$.

**Possibilistic Clustering.** To capture feature typicality and enhance robustness to outliers, we introduce a differentiable possibilistic clustering based on Possibilistic C-Means (Krishnapuram & Keller, 2002). This base clustering method relaxes the probabilistic constraint of fuzzy methods, instead measuring the possibility of a point's membership, allowing anomalous points to have low membership values across all clusters. The possibilistic assignment matrix $\boldsymbol{A} \in (0, 1)^{M \times N}$ is computed via a sigmoid function on $\boldsymbol{S} \in \mathbb{R}^{M \times N}$:

$$\boldsymbol{A} = 1/\big(1 + \exp(-\boldsymbol{S})\big). \qquad (8)$$

Here, the membership scores for a given point are independent and not constrained to sum to one. The cluster representations $\boldsymbol{C} \in \mathbb{R}^{M \times D}$ are then updated via a weighted aggregation identical to that in fuzzy clustering (Eq. 7).

**Probabilistic Clustering.** We also implement a differentiable probabilistic clustering, inspired by the Gaussian Mixture Models (Zivkovic, 2004). Compared with similarity-based methods, this base clustering method takes a generative approach to explicitly model the density and specific geometric shape (*e.g.*, variance or spread) of each cluster. Assuming a diagonal covariance, the log-likelihood $\boldsymbol{L} \in \mathbb{R}^{M \times N}_{\leq 0}$ is computed as:

$$\boldsymbol{L}_{ij} = -0.5\big(\sum_{d=1}^{D} \frac{(\boldsymbol{P}_{jd}^k - \boldsymbol{O}_{id}^k)^2}{\boldsymbol{\sigma}_{id}^2} + \sum_{d=1}^{D} \log(\boldsymbol{\sigma}_{id}^2)\big), \quad (9)$$

where $\boldsymbol{\sigma} \in \mathbb{R}^{M \times D}$ is the learnable variance matrix. We apply the same affine transformation as in Eq. 3 to get an adaptive log-likelihood $\boldsymbol{L}'$. The final assignment (or responsibility) matrix $\boldsymbol{A} \in (0, 1)^{M \times N}$ is then computed by:

$$\boldsymbol{A}_{ij} = \frac{\exp(\boldsymbol{L}'_{ij} + \log \boldsymbol{\omega}_i)}{\sum_{m=1}^{M} \exp(\boldsymbol{L}'_{mj} + \log \boldsymbol{\omega}_m)}, \qquad (10)$$

where $\boldsymbol{\omega}_m$ denotes the learnable cluster prior. The cluster representations $\boldsymbol{C} \in \mathbb{R}^{M \times D}$ are subsequently updated using the same weighted aggregation as in Eq. 7, with the responsibilities $\boldsymbol{A}$ serving as the soft weights.

**Ensemble Generation.** The base clustering methods described above define the individual components that compose our ensemble. Formally, we define this ensemble as $\Pi = \{\pi^1, \dots, \pi^T\}$. We now present how these components $\pi^t$ are jointly implemented and organized within the DEC module. Specifically, the input feature points $\boldsymbol{P} \in \mathbb{R}^{N \times D}$ are projected into $T+1$ distinct spaces: one primary space for querying and $T$ individual subspaces for clustering. Each space projection adopts a smaller channel dimension than the input, serving as a lightweight transformation for subspace diversification. This projection yields a primary feature map $\boldsymbol{P}^q \in \mathbb{R}^{N \times D}$ and $T$ pairs of *key-value* feature maps

$\{(\boldsymbol{P}^k, \boldsymbol{P}^v)^t\}_{t=1}^T$. Each pair $(\boldsymbol{P}^k, \boldsymbol{P}^v)^t$ is then processed by its corresponding base clustering $\pi^t$, applying its unique assignment formulation (*e.g.*, Eq. 4 or 6) to generate a set of cluster representations $\boldsymbol{C}^t \in \mathbb{R}^{M^t \times D}$. Collectively, this process generates the full ensemble $\Pi$, which serves as the basis for the subsequent consensus aggregation.

**Consensus Aggregation.** For these distinct cluster representations of the ensemble $\Pi$, a differentiable consensus mechanism is designed to integrate them into a unified feature representation. All clusters from the ensemble $\Pi$ are concatenated into a unified matrix:

$$\boldsymbol{C}' = \texttt{Concat}(\boldsymbol{C}^1, \dots, \boldsymbol{C}^T) \in \mathbb{R}^{M' \times D}, \qquad (11)$$

where $M' = \sum_{t=1}^{T} M^t$. This set serves as a comprehensive dictionary of learned structural patterns. The primary features $\boldsymbol{P}^q \in \mathbb{R}^{N \times D}$ query this dictionary $\boldsymbol{C}'$ to compute the adaptive similarity $\boldsymbol{S}'$ (Eq. 3), which yields the consensus assignment matrix $\boldsymbol{A}' \in (0, 1)^{M' \times N}$:

$$\boldsymbol{A}'_{ij} = \exp(\boldsymbol{S}'_{ij})/ \sum_{m=1}^{M'} \exp(\boldsymbol{S}'_{mj}). \qquad (12)$$

Here, each column of $\boldsymbol{A}'$ provides a distribution of weights, indicating how to reconstruct an input feature from the ensemble $\Pi$. The final output feature points $\boldsymbol{P}' \in \mathbb{R}^{N \times D}$ are obtained by a residual redistribution:

$$\boldsymbol{P}' = \boldsymbol{P} + \texttt{FC}(\boldsymbol{A}'^{\top} \boldsymbol{C}'), \qquad (13)$$

where $\texttt{FC}(\cdot)$ denotes a linear layer. As such, the most relevant structural information from the ensemble $\Pi$ is encoded adaptively for each feature point, yielding a refined and context-aware representation $\boldsymbol{P}'$. This completes the full DEC pipeline, where multiple clustering methods are integrated into a single coherent representation through differentiable consensus mechanism.

### 3.3. Analysis

**Why design a differentiable consensus instead of co-association matrix?** Traditional consensus functions typically rely on non-differentiable co-association matrices (Strehl & Ghosh, 2002; Domeniconi & Al-Razgan, 2009), which precludes their direct application in deep learning. In contrast, we reformulate consensus aggregation as a differentiable feature reconstruction task (see **Consensus Aggregation** in §3.2), where input features query a concatenated dictionary (Eq. 12) of learned patterns to achieve differentiable matrix computation. Furthermore, unlike classical methods that require explicit reliability estimation (Li & Ding, 2008; Huang et al., 2017), our approach implicitly learns to weight cluster contributions to emphasize more reliable cluster structures during optimization. Consequently, our consensus formulation (Eq. 13) optimizes for feature

reconstruction rather than partition consistency, better aligning the consensus objective with representation learning.

**How does consensus aggregation differ from attention-style token mixing?** Although consensus aggregation involves query-dictionary matching, its role and semantics differ from standard attention-style token mixing. In standard self-attention, queries attend to raw token features and primarily mix information among input tokens. In contrast, the dictionary in our consensus aggregation is constructed from cluster representatives produced by multiple base clustering methods. These representatives encode structural prototypes from different clustering objectives, such as hard partitioning and soft membership modeling. The aggregation process weights these cluster-level representatives and redistributes them to reconstruct point-level features. Therefore, consensus aggregation reconstructs point-level features by adaptively selecting and redistributing ensemble clustering information from this structural dictionary, rather than directly weighting point features or mixing raw tokens.

**Does the ensemble compromise efficiency?** Efficiency is an important consideration in our ensemble design. Although multiple clustering methods are employed, the total cost is controlled by proportionally reducing the channel dimension of each subspace projection (see **Ensemble Generation** in §3.2). This strategy keeps the computation comparable to that of a single clustering process. For instance, throughput analysis (see **Ensemble Components** in §4.4) indicates that the $T = 1$ setup ("Partitional"), the default $T = 2$ configuration ("Partitional + Fuzzy"), and the $T = 3$ combination ("Partitional + Fuzzy + Possibilistic") achieve 1533.1, 1476.5, and 1450.7 images per second (img/s), respectively, indicating that increasing the ensemble size introduces minimal computational overhead.

## 4. Experiment

We evaluate ENFORMER on four vision tasks: image classification (§4.1), object detection and instance segmentation (§4.2), and semantic segmentation (§4.3). We then carry out a detailed ablation study (§4.4) and visualize the learned clustering assignment maps (§4.5).

### 4.1. Experiments on Image Classification

**Dataset.** The ImageNet-1K dataset (Deng et al., 2009) is a large-scale benchmark containing 1.28 million training images and 50,000 validation images across 1,000 categories.

**Training.** We train all models for 310 epochs using the AdamW (Loshchilov & Hutter, 2017) optimizer with cyclic regularization and a learning rate of $1e-3$. The learning rate schedule consists of a 5-epoch linear warm-up, a 295-epoch cosine decay, and a 10-epoch cooldown phase. We use a batch size of 1024, a momentum of 0.9, and a weight de-

*Table 1.* Quantitative results on ImageNet-1K (Deng et al., 2009) `val` for **classification** (§4.1). Throughput is measured on a single NVIDIA V100 GPU at batch size 256 for fair comparison.

| | Method | Param (M)↓ | FLOPs (G)↓ | Throughput (img/s)↑ | Top-1 Acc (%)↑ |
|---|---|---|---|---|---|
| Conv. | ResNet18 (He et al., 2016) | 12 | 1.8 | 4284.9 | 69.8 |
| | ResNet50 (He et al., 2016) | 26 | 4.1 | 1206.0 | 79.8 |
| | ConvMixer$_{512/16}$ (Trockman & Kolter, 2022) | 5.4 | - | - | 73.8 |
| | ConvMixer$_{1024/12}$ (Trockman & Kolter, 2022) | 14.6 | - | - | 77.8 |
| | ConvMixer$_{768/32}$ (Trockman & Kolter, 2022) | 21.1 | - | 139.7 | 80.2 |
| Attention | ViT-B/16 (Dosovitskiy et al., 2020) | 86.0 | 55.5 | 86.4 | 77.9 |
| | ViT-L/16 (Dosovitskiy et al., 2020) | 307 | 190.7 | 26.6 | 76.5 |
| | PVT-Tiny (Wang et al., 2021) | 13.2 | 1.9 | - | 75.1 |
| | PVT-Small (Wang et al., 2021) | 24.5 | 3.8 | - | 79.8 |
| | Swin-Tiny (Liu et al., 2021b) | 29 | 4.5 | 631.1 | 81.3 |
| | Swin-Small (Liu et al., 2021b) | 50 | 8.7 | 374.9 | 83.0 |
| MLP | ResMLP-12 (Touvron et al., 2022) | 15.0 | 3.0 | 1499.0 | 76.6 |
| | ResMLP-24 (Touvron et al., 2022) | 30.0 | 6.0 | 741.6 | 79.4 |
| | ResMLP-36 (Touvron et al., 2022) | 45.0 | 8.9 | 484.6 | 79.7 |
| | MLP-Mixer-B/16 (Tolstikhin et al., 2021) | 59.0 | 12.7 | 387.7 | 76.4 |
| | MLP-Mixer-L/16 (Tolstikhin et al., 2021) | 207.0 | 44.8 | 114.2 | 71.8 |
| | gMLP-Ti (Liu et al., 2021a) | 6.0 | 1.4 | 1440.0 | 72.3 |
| | gMLP-S (Liu et al., 2021a) | 20.0 | 4.5 | 650.5 | 79.6 |
| Clustering | CoC-Tiny (Ma et al., 2023) | 5.3 | 1.1 | 1146.7 | 71.8 |
| | CoC-Small (Ma et al., 2023) | 14.0 | 2.8 | 852.1 | 77.5 |
| | CoC-Medium (Ma et al., 2023) | 27.9 | 5.9 | 345.7 | 81.0 |
| | FEC-Small (Chen et al., 2024) | 5.5 | 1.4 | 1042.9 | 72.7 |
| | FEC-Base (Chen et al., 2024) | 14.4 | 3.4 | 754.1 | 78.1 |
| | FEC-Large (Chen et al., 2024) | 28.3 | 6.5 | 342.1 | 81.2 |
| | **ENFORMER-Small** (Ours) | 8.1 | 1.1 | 1476.5 | **78.9** |
| | **ENFORMER-Base** (Ours) | 14.8 | 2.5 | 1075.5 | **81.2** |
| | **ENFORMER-Large** (Ours) | 29.4 | 4.8 | 621.7 | **82.6** |

cay of 0.05. Following the established training recipes (Yu et al., 2022; Chen et al., 2024) for fair comparison, we employ a suite of data augmentations, including RandAugment (Cubuk et al., 2020), Mixup (Zhang et al., 2017), CutMix (Yun et al., 2019), Random Erasing (Zhong et al., 2020), Random Horizontal Flip (Krizhevsky et al., 2012), and Label Smoothing (Szegedy et al., 2016). The input image resolution is fixed at $224 \times 224$.

**Evaluation.** In line with (Ma et al., 2023; Chen et al., 2024), all models are evaluated on a single $224 \times 224$ center crop from each validation image without test-time augmentation. All experiments are implemented in PyTorch using the timm library (Wightman, 2019), conducted on 4 NVIDIA A40 GPUs. Throughput is measured on a single NVIDIA V100 GPU with a batch size of 256.

**Metric.** The number of parameters (Param), floating-point operations (FLOPs), Top-1 accuracy (Top-1 Acc), and throughput (img/s) are reported as the primary metrics.

**Performance Comparison.** Table 1 reports classification results on ImageNet-1K (Deng et al., 2009) `val`. Compared with existing clustering-based backbones, EN-FORMER consistently achieves superior accuracy-efficiency trade-offs across all model scales. For instance, at comparable parameter counts, ENFORMER-Base/Large surpass FEC-Base/Large (Chen et al., 2024) by **3.1**% and **1.4**% Top-1 accuracy, respectively, while requiring fewer FLOPs (**2.5/4.8** *vs.* **3.4/6.5**G) and achieving higher throughput (**1075.5/621.7** *vs.* 754.1/342.1 img/s). Notably, EN-FORMER-Large runs approximately **1.8**× faster than FEC-Large. Furthermore, ENFORMER-Large also outperforms

*Table 2.* Quantitative results on COCO (Lin et al., 2014) `val2017` for **object detection** and **instance segmentation** (§4.2).

| Backbone | Param(M)↓ | $AP^{box}$ ↑ | $AP^{box}_{50}$ ↑ | $AP^{mask}$ ↑ | $AP^{mask}_{50}$ ↑ |
|---|---|---|---|---|---|
| ResNet18 (He et al., 2016) | 31.2 | 34.0 | 54.0 | 31.2 | 51.0 |
| ResNet50 (He et al., 2016) | 44.2 | 38.0 | 58.6 | 34.4 | 55.1 |
| PVT-Tiny (Wang et al., 2021) | 32.9 | 36.7 | 59.2 | 35.1 | 56.7 |
| PVT-Small (Wang et al., 2021) | 44.1 | 40.4 | 62.9 | 37.8 | 60.1 |
| CoC-Small/4 (Ma et al., 2023) | 32.7 | 35.9 | 58.3 | 33.8 | 55.3 |
| CoC-Small/25 (Ma et al., 2023) | 32.7 | 37.5 | 60.1 | 35.4 | 57.1 |
| CoC-Small/49 (Ma et al., 2023) | 32.7 | 37.2 | 59.8 | 34.9 | 56.7 |
| CoC-Medium/4 (Ma et al., 2023) | 46.7 | 38.6 | 61.1 | 36.1 | 58.2 |
| CoC-Medium/25 (Ma et al., 2023) | 46.7 | 40.1 | 62.8 | 37.4 | 59.9 |
| CoC-Medium/49 (Ma et al., 2023) | 46.7 | 40.6 | 63.3 | 37.6 | 60.1 |
| FEC-Small (Chen et al., 2024) | 24.3 | 35.6 | 57.5 | 33.6 | 54.7 |
| FEC-Base (Chen et al., 2024) | 33.1 | 37.9 | 60.1 | 35.5 | 57.2 |
| FEC-Large (Chen et al., 2024) | 47.1 | 39.9 | 62.5 | 37.3 | 59.5 |
| **ENFORMER-Small** (Ours) | 28.2 | **41.3** | **63.3** | **38.1** | **60.2** |
| **ENFORMER-Base** (Ours) | 35.0 | **42.8** | **64.6** | **39.3** | **61.7** |
| **ENFORMER-Large** (Ours) | 50.3 | **44.0** | **65.6** | **40.0** | **62.5** |

*Table 3.* Quantitative results on ADE20K (Zhou et al., 2017) `val` for **semantic segmentation** (§4.3).

| Backbone | Param(M)↓ | mIoU(%)↑ |
|---|---|---|
| ResNet18 (He et al., 2016) | 15.5 | 32.9 |
| ResNet50 (He et al., 2016) | 28.5 | 36.7 |
| PVT-Tiny (Wang et al., 2021) | 17.0 | 35.7 |
| PVT-Small (Wang et al., 2021) | 28.2 | 39.8 |
| CoC-Small/4 (Ma et al., 2023) | 17.6 | 36.6 |
| CoC-Small/25 (Ma et al., 2023) | 17.6 | 36.4 |
| CoC-Small/49 (Ma et al., 2023) | 17.6 | 36.3 |
| CoC-Medium/4 (Ma et al., 2023) | 31.5 | 40.2 |
| CoC-Medium/25 (Ma et al., 2023) | 31.5 | 40.6 |
| CoC-Medium/49 (Ma et al., 2023) | 31.5 | 40.8 |
| FEC-Small (Chen et al., 2024) | 9.1 | 35.3 |
| FEC-Base (Chen et al., 2024) | 18.0 | 37.7 |
| FEC-Large (Chen et al., 2024) | 31.9 | 40.5 |
| **ENFORMER-Small** (Ours) | 12.3 | **43.3** |
| **ENFORMER-Base** (Ours) | 19.3 | **44.3** |
| **ENFORMER-Large** (Ours) | 34.6 | **46.6** |

Swin-Tiny (Liu et al., 2021b) by **+1.3**% Top-1 accuracy, with comparable throughput. These results demonstrate the favorable balance of ENFORMER between accuracy and computational efficiency.

### 4.2. Experiments on Object Detection and Instance Segmentation

**Dataset.** The COCO 2017 dataset (Lin et al., 2014) is a common benchmark for object detection and instance segmentation. It comprises over 118K training images and 5K validation images across 80 object categories.

**Training.** Following standard protocols (Wang et al., 2021), we adopt the Mask R-CNN (He et al., 2017) framework with an FPN (Lin et al., 2017) neck, initialized with ImageNet-1K (Deng et al., 2009) pre-trained backbone weights. We use the standard $1\times$ schedule (12 epochs) in MMDetection (Chen et al., 2019), with an AdamW optimizer, an initial learning rate of $2e-4$, and a batch size of 16. Input images are resized to a maximum shape of $1333\times800$ pixels.

**Evaluation.** We evaluate the models without test-time augmentation on a single-scale input where images are resized to a maximum shape of $1333\times800$ pixels. All experiments are conducted using the MMDetection (Chen et al., 2019) toolbox on 4 NVIDIA A40 GPUs.

**Metric.** The Average Precision for boxes ($AP^{box}$, $AP^{box}_{50}$) and masks ($AP^{mask}$, $AP^{mask}_{50}$) are reported.

**Performance Comparison.** Table 2 summarizes the detection and segmentation results on COCO (Lin et al., 2014) `val2017`. Compared with other competitors, EN-FORMER consistently achieves superior detection and segmentation accuracy under similar model scales. Specifically, ENFORMER-Small/Base/Large achieve promising gains of **5.7/4.9/4.1** $AP^{box}$ and **4.5/3.8/2.7** $AP^{mask}$ against FEC-Small/Base/Large, respectively. These results demonstrate that the proposed architecture generalizes effectively to dense prediction tasks and provides stronger structural representations for instance-level understanding.

### 4.3. Experiments on Semantic Segmentation

**Dataset.** The ADE20K dataset (Zhou et al., 2017) is a challenging scene parsing benchmark with 20k training images and 2k validation images across 150 semantic categories.

**Training.** We use the Semantic FPN (Xiao et al., 2018) framework and initialize our backbones with ImageNet-1K (Deng et al., 2009) pre-trained weights. Following (Ma et al., 2023; Chen et al., 2024), we train for 80K iterations using the AdamW (Loshchilov & Hutter, 2017) optimizer with an initial learning rate of $1e-4$ and a batch size of 16. The learning rate follows a polynomial decay schedule (power$=0.9$). The input images are cropped to $512\times512$.

**Evaluation.** We evaluate the models on a $2048\times512$ resized from each validation image without test-time augmentation. All experiments are conducted using the MMSegmentation (Contributors, 2020) toolbox on 4 NVIDIA A40 GPUs.

**Metric.** Mean Intersection over Union (mIoU) is reported.

**Performance Comparison.** Table 3 reports the semantic segmentation results on ADE20K (Zhou et al., 2017) `val`. ENFORMER exhibits remarkable semantic segmentation performance compared to existing backbones. Specifically, ENFORMER-Small/Base/Large outperform FEC-Small/Base/Large (Chen et al., 2024) by **8.0/6.6/6.1** mIoU, respectively. These results demonstrate strong capability of ENFORMER in semantic segmentation and its effectiveness in capturing fine-grained spatial and semantic relationships.

### 4.4. Diagnostic Experiment

To gain more insights into ENFORMER, we conduct a series of ablation studies on ImageNet-1K (Deng et al., 2009), COCO (Lin et al., 2014), and ADE20K (Zhou et al., 2017), all based on the ENFORMER-Small variant.

**Ensemble Components.** We analyze the individual components in Table 4. Based on (Chen et al., 2024), we construct a partitional clustering baseline scaled to 7.6M parameters,

*Table 4.* Ablation study of the ensemble components on ImageNet-1K (Deng et al., 2009). The "*w/o* ensemble" indicates the partitional clustering baseline without ensemble framework. **Bold** values denote our final ensemble combination.

| Ensemble Components | Param (M)↓ | Throughput (img/s)↑ | Top-1 Acc (%)↑ | Memory (GB)↓ Training | Inference |
|---|---|---|---|---|---|
| Baseline (*w/o* ensemble) | 7.6 | 1745.7 | 77.0 | 48.00 | 5.58 |
| Partitional | 7.8 | 1533.1 | 77.6 | 51.54 | 6.16 |
| Fuzzy | 7.8 | 1566.3 | 78.0 | 50.76 | 6.16 |
| **Partitional + Fuzzy** | **8.1** | **1476.5** | **78.9** | **57.51** | **6.16** |
| Partitional + Possibilistic | 8.1 | 1466.8 | 78.6 | 55.92 | 6.16 |
| Partitional + Probabilistic | 8.1 | 1377.2 | 78.8 | 70.38 | 6.16 |
| Partitional + Fuzzy + Possibilistic | 8.1 | 1450.7 | 78.5 | 57.74 | 6.16 |
| Partitional + Fuzzy + Probabilistic | 8.1 | 1332.0 | 78.9 | 68.85 | 6.16 |
| all four base clustering methods | 8.3 | 1306.8 | 78.6 | 69.56 | 6.16 |

*Table 5.* Diversity analysis of ensemble combinations.

| Method | Top-1 Acc (%)↑ | Aligned soft-JS (%)↑ |
|---|---|---|
| **Partitional + Fuzzy** | 78.9 | **0.45** |
| Partitional + Fuzzy + Probabilistic | 78.9 | 0.36 |
| all four base clustering methods | 78.6 | 0.19 |

*Table 6.* Comparison of aggregation methods.

| Aggregation Method | Param (M)↓ | FLOPs (G)↓ | Top-1 Acc (%)↑ |
|---|---|---|---|
| Linear Projection | 7.8 | 1.1 | 78.1 |
| **Consensus Aggregation** | 8.1 | 1.1 | **78.9** |

to mitigate representational bottlenecks from narrow feature subspaces in small-scale ENFORMER. This baseline achieves 77.0% Top-1 accuracy. Integrating it into our ensemble framework (as a $T = 1$ ensemble) raises the accuracy to **77.6**%, while the "Fuzzy" component performs better (**78.0**% Top-1 accuracy).

Our analysis primarily focuses on heterogeneous combinations of the four base clustering methods. This validates our central claim that *integrating multiple clustering perspectives provides a more robust representation than any single one*. We observe that peak performance is achieved at $T = 2$ and $T = 3$, with both our default "Partitional + Fuzzy" and the "Partitional + Fuzzy + Probabilistic" combinations reaching the highest accuracy of **78.9**%. This performance plateau at $T = 4$ does not imply that larger ensembles are ineffective. Rather, it suggests a nuanced trade-off: integrating too many highly heterogeneous mechanisms may introduce *conflicting structural biases* that "check" each other, rather than providing purely complementary information. This highlights that an optimal ensemble requires a careful *balance of diversity and compatibility*.

**Computational and Memory Overhead.** Table 4 also reports throughput and peak memory with batch size 1024. Overall, the overhead generally increases with the ensemble size, as reflected by lower throughput and higher training memory. However, the practical cost is determined not only by the number of components, but also by the instantiated clustering type. For example, at $T = 2$, "Partitional + Probabilistic" is slower and more memory-intensive than "Partitional + Fuzzy" and "Partitional + Possibilistic" (1377.2 *vs.* 1476.5/1466.8 img/s; 70.38 *vs.* 57.51/55.92 GB). A similar trend appears at $T = 3$, where "Partitional + Fuzzy + Probabilistic" is slower than "Partitional + Fuzzy + Possibilistic" (1332.0 *vs.* 1450.7 img/s).

Meanwhile, the scaling is moderated by the shared feature budget. Increasing $T$ reduces the per-component feature dimension, which partially offsets the cost of adding more components. For instance, from "Partitional + Probabilistic" to "Partitional + Fuzzy + Probabilistic", the throughput decreases from 1377.2 to 1332.0 img/s, while training memory slightly decreases from 70.38 to 68.85 GB. This suggests that the total dictionary size and the clustering type jointly determine the practical overhead. In addition, inference memory remains nearly unchanged across different ensemble combinations. Appendix A.1 provides a theoretical analysis of the computational and memory complexity of different clustering components.

**Component Diversity.** To further examine how the size of a heterogeneous ensemble relates to component diversity, we introduce aligned soft-JS (see Appendix D.3 for detailed definition) to measure the discrepancy among soft assignment behaviors across clustering components. A higher aligned soft-JS value indicates more diverse clustering perspectives, while a lower value suggests stronger redundancy. As shown in Table 5, "Partitional + Fuzzy" and "Partitional + Fuzzy + Probabilistic" achieve the best Top-1 accuracy of **78.9**% with relatively high aligned soft-JS values of 0.45 and 0.36, respectively. In contrast, the all-four setting has a lower aligned soft-JS value of 0.19 and also a slightly lower accuracy of **78.6**%. This trend suggests that the benefit of ensemble clustering depends more on complementary assignment behaviors than on simply increasing the number of components.

**Aggregation Method.** We compare the proposed consensus aggregation with a simple aggregation baseline. In this baseline, the cluster outputs from different base clustering methods are concatenated and directly projected by a linear layer to obtain the aggregated representation. As shown in Table 6, consensus aggregation improves Top-1 accuracy by **0.8**% over linear projection. Compared with simple linear projection, consensus aggregation can more effectively

*Table 7.* Ablation study on the number of clusters per stage ($s$). The parameter count is consistent across all settings. Note that cluster numbers are simplified for brevity (*e.g.*, $4^2$ denotes a $4 \times 4$ output resolution of the `AdaPool` operation). **Bold**, red, and blue values highlight the final configurations used for our all-scale models on ImageNet-1K (Deng et al., 2009) **classification**, COCO (Lin et al., 2014) object detection and instance segmentation, and ADE20K (Zhou et al., 2017) semantic segmentation, respectively.

| Partitional | | | | Fuzzy | | | | ImageNet-1K |
|---|---|---|---|---|---|---|---|---|
| $s=1$ | $s=2$ | $s=3$ | $s=4$ | $s=1$ | $s=2$ | $s=3$ | $s=4$ | Top-1(%)↑ |
| $4^2$ | $4^2$ | $2^2$ | $2^2$ | $5^2$ | $4^2$ | $3^2$ | $2^2$ | 78.6 |
| $5^2$ | $4^2$ | $3^2$ | $2^2$ | $5^2$ | $4^2$ | $3^2$ | $2^2$ | 78.6 |
| **$5^2$** | **$4^2$** | **$3^2$** | **$2^2$** | **$6^2$** | **$5^2$** | **$4^2$** | **$3^2$** | **78.9** |
| $6^2$ | $5^2$ | $4^2$ | $3^2$ | $6^2$ | $5^2$ | $4^2$ | $3^2$ | 78.6 |

*(a)* ImageNet-1K **classification** ablation results

| Partitional | | | | Fuzzy | | | | COCO | | ADE20K |
|---|---|---|---|---|---|---|---|---|---|---|
| $s=1$ | $s=2$ | $s=3$ | $s=4$ | $s=1$ | $s=2$ | $s=3$ | $s=4$ | AP$^{\text{box}}$ ↑ | AP$^{\text{mask}}$ ↑ | mIoU(%)↑ |
| $5^2$ | $4^2$ | $3^2$ | $2^2$ | $6^2$ | $5^2$ | $4^2$ | $3^2$ | 40.6 | 37.4 | 41.8 |
| $10^2$ | $10^2$ | $10^2$ | $10^2$ | $10^2$ | $10^2$ | $10^2$ | $10^2$ | 41.1 | 38.0 | 42.3 |
| $10^2$ | $10^2$ | $10^2$ | $10^2$ | $12^2$ | $12^2$ | $12^2$ | $12^2$ | 41.3 | 38.1 | 42.5 |
| $12^2$ | $12^2$ | $12^2$ | $12^2$ | $12^2$ | $12^2$ | $12^2$ | $12^2$ | 41.1 | 38.1 | 43.3 |
| $12^2$ | $12^2$ | $12^2$ | $12^2$ | $14^2$ | $14^2$ | $14^2$ | $14^2$ | 41.2 | 38.1 | 42.4 |

*(b)* Downstream task ablation results (COCO & ADE20K).

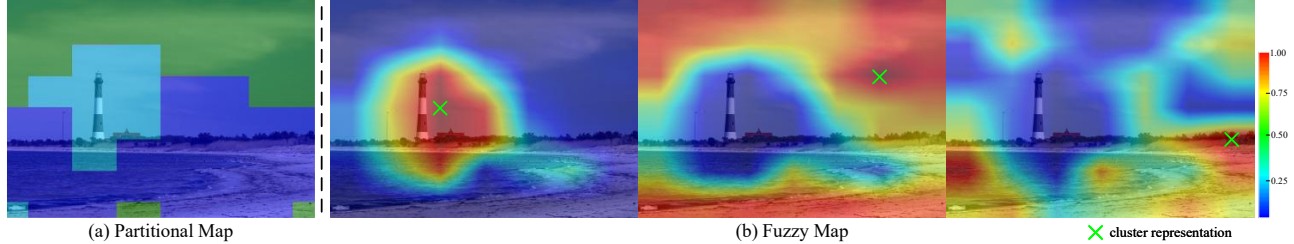

(a) Partitional Map  (b) Fuzzy Map  ✕ cluster representation

*Figure 3.* Visualization of clustering assignment maps. (a) The Partitional Map illustrates *hard*, non-overlapping assignments, where regions with the same color belong to a single, discrete cluster. (b) The Fuzzy Maps visualize the *soft*, continuous membership degrees (red=high) for three representative cluster representations. The green 'X' marks the cluster representation, highlighting how different representations specialize on the main object (*e.g.*, the lighthouse) versus the background context (*e.g.*, the sky). See §4.5 for details.

refine features by adaptively selecting and redistributing structural information from the learned cluster dictionary.

**Cluster Number Analysis.** We then analyze the impact of cluster numbers on ImageNet (Deng et al., 2009) classification (Table 7a) for our "Partitional + Fuzzy" combination, ablating the cluster counts for each method per stage. We find that the configuration, using $[5^2, 4^2, 3^2, 2^2]$ clusters for "Partitional" and $[6^2, 5^2, 4^2, 3^2]$ for "Fuzzy" (in the $3^{rd}$ row), achieves the optimal Top-1 accuracy of **78.9**%. We thus adopt this configuration as the default for classification.

For downstream tasks (Table 7b), which require finer-grained spatial representations, more clusters are beneficial (1ˢᵗ *vs.* 2ⁿᵈ∼5ᵗʰ rows). For COCO (Lin et al., 2014), the 3ʳᵈ row configuration, using a fixed $10^2$ schedule for "Partitional" and a $12^2$ schedule for "Fuzzy", yields the best results of **41.3** AP$^{\text{box}}$ and **38.1** AP$^{\text{mask}}$. For ADE20K (Zhou et al., 2017), 4ᵗʰ row configuration, which uses a uniform fixed $12^2$ schedule for both methods, achieves the peak segmentation performance of **43.3** mIoU. We select these configurations for our final downstream experiments.

**4.5. Visualization of Clustering**

Fig. 3 depicts representative clustering assignment maps from our base clusterings, generated from the model's final stage. For clear visualization, these maps are upsampled to the original image resolution using bilinear interpolation. More visualizations are provided in the Appendix §D.5.

Our partitional clustering method provides the model with a *definitive* and *discrete* structural representation. Fig. 3(a) visualizes this: the method enforces a strict, one-to-one assignment, which is illustrated in the Partitional Map by the hard-edged, non-overlapping colored regions. This partitioning is effective at isolating the main semantic object (*e.g.*, the lighthouse) from its surroundings.

In contrast, our fuzzy clustering method offers a *comprehensive* and *continuous* structural representation. Fig. 3(b) shows several Fuzzy Maps, where soft assignments are rendered as continuous heatmaps (red=high) rather than hard boundaries, allowing each cluster to capture broad, overlapping features. For instance, different clusters (marked by a green 'X') may develop specialized focus regions while simultaneously attending to their related context.

## 5. Conclusion and Discussion

This work addresses the limited representational capacity of existing clustering-based visual backbones that rely on one single clustering algorithm. We propose ENFORMER, a new vision backbone architecture that, for the first time, successfully embeds ensemble clustering directly into the feature extraction process. Extensive experiments demonstrate that ENFORMER consistently outperforms existing clustering-based counterparts, establishing a new standard for this paradigm in both accuracy and computational efficiency. A detailed discussion of limitations and future work is provided in the Appendix §F. We hope this work can encourage further exploration of ensemble clustering as a foundation for advancing visual representation learning.

## Acknowledgements

This work was supported by the National Natural Science Foundation of China (No. T25B2005, 62506169, 62472222, 62302217), Natural Science Foundation of Jiangsu Province (No. BK20240080), Jiangsu Provincial Scientific Research Center of Applied Mathematics (No. BK20233002), National Defense Science and Technology Industry Bureau Technology Infrastructure Project (No. JSZL2024606C001), and CCF-Tencent Open Fund.

## Impact Statement

This work develops a deep ensemble clustering framework that learns visual representations by integrating multiple clustering perspectives to mitigate the inherent bias of individual algorithms. On the positive side, our approach has the potential to benefit a wide variety of real-world applications, such as autonomous driving, medical image diagnosis, and robotic perception. Furthermore, the high computational efficiency of ENFORMER contributes to the goal of "Green AI," significantly reducing the energy consumption and carbon footprint associated with large-scale visual recognition tasks. Regarding potential negative impacts, as with any general-purpose vision backbone, our model could theoretically be employed for surveillance or privacy-intrusive applications. In addition, since our method relies on clustering latent features, it may inherit or even amplify data biases present in pre-training datasets (*e.g.*, ImageNet (Deng et al., 2009)), potentially leading to biased representations in sensitive contexts. We advocate that future deployment should include rigorous bias testing and ethical auditing.

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

For a better understanding of the main paper, we provide additional details in this appendix, which is organized as follows:

- §A provides more analysis.

- §B reports more experimental details.

- §C introduces more experimental results.

- §D offers more diagnostic experiments.

- §E presents the architecture and pseudo code of ENFORMER.

- §F discusses our limitations and future work.

## A. More Analysis

### A.1. Complexity Analysis of Components

We analyze the dominant computational and memory costs of the base clustering components used in ENFORMER. Let $N$ denote the number of feature points, $D$ the input channel dimension, $d$ the projected subspace dimension of each clustering component, $M^t$ the number of cluster representatives in the $t$-th component, and $T$ the number of ensemble components. For the sake of simplicity, we focus on the computational and memory complexity of the dominant operations.

For each similarity-based base clustering component (*i.e.*, partitional, fuzzy and possibilistic clustering), the dominant operations include assignment computation between $N$ feature points and $M^t$ cluster representatives, followed by weighted aggregation from points to cluster representatives. The assignment computation requires $\mathcal{O}(M^t N d)$ time and stores an assignment map with $\mathcal{O}(M^t N)$ activation memory. The subsequent weighted aggregation also requires $\mathcal{O}(M^t N d)$ time while reusing the same assignment map for point-to-cluster aggregation. Thus, the dominant complexity of each similarity-based component is $\mathcal{O}(M^t N d)$ time with $\mathcal{O}(M^t N)$ activation memory. Probabilistic clustering has the same dominant time complexity of $\mathcal{O}(M^t N d)$, but requires $\mathcal{O}(M^t N d)$ activation memory complexity, since it additionally introduces feature-wise intermediate tensors for element-wise distance, variance, and likelihood computations.

For an ensemble with $T$ components, the dominant time complexity of ensemble generation is therefore $\mathcal{O}(\sum_{t=1}^{T} \kappa_t M^t N d)$ with dominant activation memory complexity $\mathcal{O}(\sum_{t \in \mathcal{S}} M^t N + \sum_{t \in \mathcal{P}} M^t N d)$, where $\mathcal{S}$ denotes the set of similarity-based components and $\mathcal{P}$ denotes the set of probabilistic components. The first term mainly comes from assignment maps in similarity-based clustering, while the second term accounts for feature-wise intermediate tensors introduced by probabilistic likelihood computation.

### A.2. Complexity Analysis of Aggregation

We further analyze the complexity of the proposed consensus aggregation. Let $M'$ denote the total number of cluster representatives after concatenation, and $D$ denote the output channel dimension.

In consensus aggregation, the cluster representatives from all base clustering components are concatenated into a unified dictionary of size $M'$. The primary features then query this dictionary to compute the consensus assignment matrix. This query-to-dictionary matching requires $\mathcal{O}(M' N d)$ time and $\mathcal{O}(M' N)$ activation memory. The subsequent redistribution from the cluster dictionary back to point-level features also requires $\mathcal{O}(M' N d)$ time. Finally, the output linear projection contributes $\mathcal{O}(N d D)$ time.

Therefore, the overall time complexity of consensus aggregation is $\mathcal{O}(M' N d + N d D)$, with dominant activation memory complexity $\mathcal{O}(M' N)$. This analysis shows that the aggregation cost scales primarily with the total dictionary size $M'$ and the projected subspace dimension $d$. Together with the shared feature budget in ensemble generation, this explains why ENFORMER can incorporate multiple clustering components while maintaining a favorable efficiency–accuracy trade-off.

## B. More Experimental Detail

**Image Classification Setup.** We implement ENFORMER using the timm library (Wightman, 2019) and train all models with a fixed random seed of 42. For the ENFORMER-Small, Base, and Large variants, we consistently employ a stochastic depth (drop path) rate of 0.1. Following prior works (Liu et al., 2021b; Chen et al., 2024), we measure FLOPs and throughput

using standard evaluation scripts. Specifically, the reported throughput is averaged over 100 iterations following 20 warm-up iterations to ensure stability.

**Downstream Tasks.** For object detection, we adopt the standard $1\times$ schedule (12 epochs), using a Step LR scheduler that decays the learning rate by 0.1 at epochs 8 and 11. For semantic segmentation, we train for 80K iterations using a Polynomial LR scheduler. All parameter counts are calculated using the official scripts provided by these frameworks (Chen et al., 2019; Contributors, 2020).

## C. More Experimental Results

### C.1. Quantitative Result on ImageNet-1K

Table 8 provides an extended comparison on ImageNet-1K classification. In addition to the methods reported in Table 1, we include representative recent vision backbones from convolutional, state-space, graph-based, and clustering-based architectures. We also report throughput on NVIDIA RTX 4090 and A40 GPUs under the same single-GPU evaluation setting with batch size 256. The results further confirm the favorable accuracy-efficiency trade-off of ENFORMER. Moreover, Table 8 reports results over three repeated runs, where the performance deviations are small.

*Table 8.* Quantitative results on ImageNet-1K (Deng et al., 2009) `val` for **classification** (§C.1).

| | Method | Param (M)↓ | FLOPs (G)↓ | Throughput (img/s)↑ | | | Top-1 Acc (%)↑ |
|---|---|---|---|---|---|---|---|
| | | | | V100 | 4090 | A40 | |
| **Conv.** | ResNet18 (He et al., 2016) | 12 | 1.8 | 4284.9 | 6828.8 | 5152.7 | 69.8 |
| | ResNet50 (He et al., 2016) | 26 | 4.1 | 1206.0 | 2144.2 | 1528.6 | 79.8 |
| | ConvMixer$_{512/16}$ (Trockman & Kolter, 2022) | 5.4 | - | - | 858.4 | 520.5 | 73.8 |
| | ConvMixer$_{1024/12}$ (Trockman & Kolter, 2022) | 14.6 | - | - | 496.8 | 311.3 | 77.8 |
| | ConvMixer$_{768/32}$ (Trockman & Kolter, 2022) | 21.1 | - | 139.7 | 1402.1 | 795.5 | 80.2 |
| | ConvNeur-M1 (Yang et al., 2026) | 4.3 | 0.7 | - | 1482.1 | - | 75.4 |
| | ConvNeur-M2 (Yang et al., 2026) | 6.1 | 1.0 | - | 1167.9 | - | 77.6 |
| | ConvNeur-M3 (Yang et al., 2026) | 10.6 | 1.8 | - | 779.1 | - | 80.0 |
| | ConvNeur-M4 (Yang et al., 2026) | 18.1 | 3.1 | - | 522.5 | - | 81.5 |
| **Attention** | ViT-B/16 (Dosovitskiy et al., 2020) | 86.0 | 55.5 | 86.4 | 836.0 | 383.9 | 77.9 |
| | ViT-L/16 (Dosovitskiy et al., 2020) | 307 | 190.7 | 26.6 | 262.0 | 116.5 | 76.5 |
| | PVT-Tiny (Wang et al., 2021) | 13.2 | 1.9 | - | 3062.6 | 1824.2 | 75.1 |
| | PVT-Small (Wang et al., 2021) | 24.5 | 3.8 | - | 1706.2 | 1001.7 | 79.8 |
| | Swin-Tiny (Liu et al., 2021b) | 29 | 4.5 | 631.1 | 1499.0 | 833.0 | 81.3 |
| | Swin-Small (Liu et al., 2021b) | 50 | 8.7 | 374.9 | 889.0 | 485.2 | 83.0 |
| **MLP** | ResMLP-12 (Touvron et al., 2022) | 15.0 | 3.0 | 1499.0 | 3226.0 | 1835.8 | 76.6 |
| | ResMLP-24 (Touvron et al., 2022) | 30.0 | 6.0 | 741.6 | 1635.7 | 912.2 | 79.4 |
| | ResMLP-36 (Touvron et al., 2022) | 45.0 | 8.9 | 484.6 | 1089.4 | 602.2 | 79.7 |
| | MLP-Mixer-B/16 (Tolstikhin et al., 2021) | 59.0 | 12.7 | 387.7 | 1105.3 | 544.5 | 76.4 |
| | MLP-Mixer-L/16 (Tolstikhin et al., 2021) | 207.0 | 44.8 | 114.2 | 341.8 | 161.2 | 71.8 |
| | gMLP-Ti (Liu et al., 2021a) | 6.0 | 1.4 | 1440.0 | 3728.9 | 2280.9 | 72.3 |
| | gMLP-S (Liu et al., 2021a) | 20.0 | 4.5 | 650.5 | 1637.1 | 956.3 | 79.6 |
| **SSM** | Mamba$^{\circledR}$-T (Wang et al., 2025) | 9.0 | 5.1 | - | - | - | 77.4 |
| | Mamba$^{\circledR}$-S (Wang et al., 2025) | 28.0 | 9.9 | - | - | - | 81.1 |
| **GNN** | ViG-Ti (Han et al., 2022) | 7.1 | 1.3 | - | 1543.4 | - | 73.9 |
| | ViG-S (Han et al., 2022) | 22.7 | 4.5 | - | 731.2 | - | 80.4 |
| **Clustering** | CoC-Tiny (Ma et al., 2023) | 5.3 | 1.1 | 1146.7 | 1784.8 | 953.8 | 71.8 |
| | CoC-Small (Ma et al., 2023) | 14.0 | 2.8 | 852.1 | 1452.5 | 808.8 | 77.5 |
| | CoC-Medium (Ma et al., 2023) | 27.9 | 5.9 | 345.7 | 599.5 | 328.4 | 81.0 |
| | FEC-Small (Chen et al., 2024) | 5.5 | 1.4 | 1042.9 | 1459.1 | 935.0 | 72.7 |
| | FEC-Base (Chen et al., 2024) | 14.4 | 3.4 | 754.1 | 1201.8 | 768.1 | 78.1 |
| | FEC-Large (Chen et al., 2024) | 28.3 | 6.5 | 342.1 | 545.4 | 352.1 | 81.2 |
| | ClusterFormer-Tiny (Liang et al., 2023) | 27.9 | - | - | - | - | 81.3 |
| | ClusterFormer-Small (Liang et al., 2023) | 48.7 | - | - | - | - | 83.4 |
| | **ENFORMER-Small** (Ours) | 8.1 | 1.1 | 1476.5 | 2735.2 | 1465.2 | **78.97**$_{\pm 0.06}$ |
| | **ENFORMER-Base** (Ours) | 14.8 | 2.5 | 1075.5 | 1960.9 | 1098.9 | **81.17**$_{\pm 0.06}$ |
| | **ENFORMER-Large** (Ours) | 29.4 | 4.8 | 621.7 | 1201.9 | 645.4 | **82.60**$_{\pm 0.10}$ |

## C.2. Quantitative Result on MS COCO

Table 9 presents detailed quantitative results on the COCO (Lin et al., 2014) `val2017` dataset for object detection and instance segmentation, including $AP_{75}^{box}$ and $AP_{75}^{mask}$. ENFORMER consistently outperforms existing clustering-based backbones across all model scales, demonstrating its superior capability in these downstream tasks.

In addition, Table 9 reports latency, training memory, and inference memory under consistent settings (8 GPUs with a total batch size of 16 on NVIDIA A40). Compared with CoC-Medium/49 and FEC-Large, ENFORMER-Large substantially reduces latency and memory usage (**122.1** *vs.* 220.1/199.7 ms/img latency, **155.60** *vs.* 284.66/292.58 GB training memory, and **0.45** *vs.* 2.90/2.95 GB inference memory). These results indicate that ENFORMER is more efficient than prior clustering-based backbones in detection and instance segmentation.

*Table 9.* Quantitative results on COCO (Lin et al., 2014) `val2017` for **object detection** and **instance segmentation** (§C.2).

| Backbone | #Param (M)↓ | $AP^{box}$ ↑ | $AP_{50}^{box}$ ↑ | $AP_{75}^{box}$ ↑ | $AP^{mask}$ ↑ | $AP_{50}^{mask}$ ↑ | $AP_{75}^{mask}$ ↑ | Latency (ms/img)↓ | Train Mem (GB)↓ | Infer Mem (GB)↓ |
|---|---|---|---|---|---|---|---|---|---|---|
| ResNet18 (He et al., 2016) | 31.2 | 34.0 | 54.0 | 36.7 | 31.2 | 51.0 | 32.7 | 22.5 | 41.28 | 0.38 |
| ResNet50 (He et al., 2016) | 44.2 | 38.0 | 58.6 | 41.4 | 34.4 | 55.1 | 36.7 | 24.3 | 55.81 | 0.52 |
| PVT-Tiny (Wang et al., 2021) | 32.9 | 36.7 | 59.2 | 39.3 | 35.1 | 56.7 | 37.3 | 39.7 | 91.74 | 0.73 |
| PVT-Small (Wang et al., 2021) | 44.1 | 40.4 | 62.9 | 43.8 | 37.8 | 60.1 | 40.3 | 58.9 | 129.95 | 0.77 |
| CoC-Small/4 (Ma et al., 2023) | 32.7 | 35.9 | 58.3 | 38.3 | 33.8 | 55.3 | 35.8 | 82.9 | 78.35 | 0.64 |
| CoC-Small/25 (Ma et al., 2023) | 32.7 | 37.5 | 60.1 | 40.0 | 35.4 | 57.1 | 37.9 | 94.1 | 124.88 | 1.14 |
| CoC-Small/49 (Ma et al., 2023) | 32.7 | 37.2 | 59.8 | 39.7 | 34.9 | 56.7 | 37.0 | 113.3 | 181.67 | 1.96 |
| CoC-Medium/4 (Ma et al., 2023) | 46.7 | 38.6 | 61.1 | 41.5 | 36.1 | 58.2 | 38.0 | 137.4 | 123.67 | 0.68 |
| CoC-Medium/25 (Ma et al., 2023) | 46.7 | 40.1 | 62.8 | 43.6 | 37.4 | 59.9 | 40.0 | 176.1 | 168.83 | 1.67 |
| CoC-Medium/49 (Ma et al., 2023) | 46.7 | 40.6 | 63.3 | 43.9 | 37.6 | 60.1 | 39.9 | 220.1 | 284.66 | 2.90 |
| FEC-Small (Chen et al., 2024) | 24.3 | 35.6 | 57.5 | 38.2 | 33.6 | 54.7 | 35.7 | 112.5 | 138.90 | 2.31 |
| FEC-Base (Chen et al., 2024) | 33.1 | 37.9 | 60.1 | 40.8 | 35.5 | 57.2 | 37.8 | 122.1 | 185.57 | 2.44 |
| FEC-Large (Chen et al., 2024) | 47.1 | 39.9 | 62.5 | 43.2 | 37.3 | 59.5 | 39.5 | 199.7 | 292.58 | 2.95 |
| **ENFORMER-Small** (Ours) | 28.2 | **41.3** | **63.3** | **44.7** | **38.1** | **60.2** | **40.9** | 66.7 | 92.53 | 0.35 |
| **ENFORMER-Base** (Ours) | 35.0 | **42.8** | **64.6** | **46.9** | **39.3** | **61.7** | **42.2** | 71.3 | 118.20 | 0.39 |
| **ENFORMER-Large** (Ours) | 50.3 | **44.0** | **65.6** | **47.7** | **40.0** | **62.5** | **43.0** | 122.1 | 155.60 | 0.45 |

## C.3. Quantitative Result on ADE20K

Table 10 further reports latency, training memory, and inference memory on the ADE20K dataset under consistent settings (8 GPUs with a total batch size of 16 on NVIDIA A40) Compared with CoC-Medium/49 and FEC-Large, ENFORMER-Large achieves lower computational cost (**67.57** *vs.* 86.06/87.64 ms/img latency, **54.81** *vs.* 68.38/81.11 GB training memory, and **0.57** *vs.* 0.79/0.95 GB inference memory). These results demonstrate that ENFORMER provides a more efficient segmentation backbone than prior clustering-based methods.

*Table 10.* Quantitative results on ADE20K for semantic segmentation.

| Backbone | Param (M)↓ | mIoU (%)↑ | Latency (ms/img)↓ | Training Mem (GB)↓ | Inference Mem (GB)↓ |
|---|---|---|---|---|---|
| CoC-Small/49 (Ma et al., 2023) | 17.6 | 36.3 | 40.26 | 43.19 | 0.52 |
| CoC-Medium/49 (Ma et al., 2023) | 31.5 | 40.8 | 86.06 | 68.38 | 0.79 |
| FEC-Small (Chen et al., 2024) | 9.1 | 35.3 | 42.12 | 47.61 | 0.71 |
| FEC-Base (Chen et al., 2024) | 18.0 | 37.7 | 45.23 | 53.13 | 0.95 |
| FEC-Large (Chen et al., 2024) | 31.9 | 40.5 | 87.64 | 81.11 | 0.95 |
| **ENFORMER-Small** (Ours) | 12.3 | **43.3** | 19.38 | 35.72 | 0.49 |
| **ENFORMER-Base** (Ours) | 19.3 | **44.3** | 43.35 | 40.45 | 0.52 |
| **ENFORMER-Large** (Ours) | 34.6 | **46.6** | 67.57 | 54.81 | 0.57 |

## D. More Diagnostic Experiments

### D.1. Additional Ablation Study of Ensemble Components

Table 4 shows that "Partitional + Fuzzy" achieves the best performance among the evaluated ensemble combinations. We further investigate homogeneous variants of these two base clustering methods to examine whether the gain mainly comes from adding more clustering branches, or whether the diversity of clustering perspectives also contributes to the improvement. As shown in Table 11, homogeneous ensembles ("Partitional + Partitional" and "Fuzzy + Fuzzy") improve over their corresponding single-component models, indicating that increasing the number of components can be beneficial. However, under the same parameter count, they still underperform the default "Partitional + Fuzzy" configuration, which

achieves **78.9**% Top-1 accuracy compared with 78.5% and 78.7% for the two homogeneous variants. This comparison suggests that the gain of the default configuration is not merely due to ensemble size, but also benefits from combining complementary clustering perspectives within the ensemble.

Regarding more complex hybrid combinations ($T = 3$), we observe that adding a second "Fuzzy" component to the optimal pair maintains the peak accuracy of 78.9% but yields no further improvement (4[th] row), while the "Probabilistic + Partitional + Partitional" combination results in a slight performance drop. We hypothesize that this plateau is partially attributed to model scale constraints (Small scale, $\approx$8M parameters). To maintain efficiency, our framework proportionally reduces the channel dimensions of each subspace. Consequently, increasing $T$ excessively in lightweight models may dilute the representational capacity of individual mechanisms. Therefore, while $T = 2$ serves as an optimal balance for these efficient models, larger ensembles may potentially unlock greater benefits in larger-capacity regimes where individual components are less constrained.

*Table 11.* Additional ablation study of the ensemble components on ImageNet-1K (Deng et al., 2009). **Bold** values denote the final combination used in our main results (see **Ensemble Components.** in §4.4).

| Ensemble Components | Param (M)↓ | Throughput (img/s)↑ | Top-1 Acc (%)↑ |
|---|---|---|---|
| **Partitional + Fuzzy** | 8.1 | 1476.5 | **78.9** |
| Partitional + Partitional | 8.1 | 1468.5 | 78.5 |
| Fuzzy + Fuzzy | 8.1 | 1536.1 | 78.7 |
| Fuzzy + Fuzzy + Partitional | 8.1 | 1469.0 | 78.9 |
| Probabilistic + Partitional + Partitional | 8.1 | 1295.1 | 78.7 |

## D.2. Sensitivity to Feature Subspace Allocation.

We analyze the sensitivity of ENFORMER to the allocation of feature subspaces across ensemble components. Since Eq. 11 aggregates component outputs in a shared subspace, varying per-component dimensions requires projection alignment before aggregation. As shown in Table 12, the balanced split (20/20) achieves the best Top-1 accuracy of **78.9**%, while moderate asymmetric allocations (16/24 and 24/16) remain competitive at 78.4%. These results suggest that ENFORMER is relatively robust to moderate subspace allocation changes, although balanced allocations provide the best trade-off under the current setting.

*Table 12.* Sensitivity analysis of feature subspace allocation on ImageNet-1K (Deng et al., 2009).

| Subspace Dimensions (Partitional / Fuzzy) | Top-1 Acc (%)↑ |
|---|---|
| 16 / 24 | 78.4 |
| 24 / 16 | 78.4 |
| **20 / 20** | **78.9** |

## D.3. Post-hoc Explanation Analysis

**Aligned Soft-JS.** We design the aligned soft-JS to measure the discrepancy between the soft assignments of two clustering components across all feature points. For each component, the per-point assignment map is normalized and converted into cluster signatures. Clusters are then optimally aligned using the Hungarian algorithm to resolve label switching. The mean Jensen-Shannon divergence over all aligned points defines the aligned soft-JS. Higher values indicate more diverse and complementary component assignments, while lower values indicate redundancy. In §4.4, we analyze the relationship between aligned soft-JS and Top-1 accuracy.

*Table 13.* Winner ratio of ensemble components on ImageNet-1K (Deng et al., 2009).

| Method | Top-1 (%) | Partitional | Fuzzy | Probabilistic | Possibilistic |
|---|---|---|---|---|---|
| Partitional + Fuzzy | 78.9 | 0.961 | 0.039 | - | - |
| Partitional + Fuzzy + Probabilistic | 78.9 | 0.772 | 0.148 | 0.079 | - |
| all four base clustering methods | 78.6 | 0.727 | 0.022 | 0.108 | 0.143 |

**Winner Ratio.** The winner ratio of a component is defined as the fraction of query points whose most similar cluster representative in the final ensemble dictionary originates from that component. This metric is designed to assess the relative contribution of each base clustering component to the ensemble output near the final decision layer. While Partitional

typically dominates, Fuzzy and Probabilistic contribute non-zero fractions. Table 13 reports these ratios for several heterogeneous combinations.

## D.4. Alternative MoE-style Solution

We additionally compare ENFORMER with a minimal MoE-style variant. Specifically, we replace the original integration-based ensemble module with an adaptive selection mechanism while keeping a comparable model scale for fair comparison. As shown in Table 14, the MoE-style variant achieves 80.5% Top-1 accuracy, which is 0.7% lower than ENFORMER-Base. This result suggests that, under the current design, integrating multiple clustering representations through consensus aggregation is more effective than adaptively selecting among them. Meanwhile, the competitive performance of the MoE-style variant indicates that dynamic routing remains a promising direction for future extensions.

*Table 14.* Comparison with a minimal MoE-style variant on ImageNet-1K (Deng et al., 2009).

| Method | Param (M)↓ | FLOPs (G)↓ | Top-1 Acc (%)↑ |
|---|---|---|---|
| MoE Clustering | 14.6 | 2.2 | 80.5 |
| **ENFORMER-Base** | 14.8 | 2.5 | **81.2** |

## D.5. Additional Visualization Results of Clustering

To provide a more comprehensive perspective on the clustering outcomes, we present extended visualization results in Figs. 4∼6. Specifically, for the fuzzy assignment maps, a green 'X' marker is employed to denote the spatial location of the feature point $p_i$ that achieves the maximum similarity score $S_{ij}$ for the $j$-th cluster center $o_j$. Formally, for each cluster representation $c_j$, the marker is positioned at $\arg\max_i(S_{ij})$. This visualization highlights the "prototypical" pixel that the cluster representation most strongly represents, thereby clarifying the semantic focus of individual clusters.

**Visualization with Different Interpolation Methods.** Fig. 4 compares the visualization effects of nearest-neighbor upsampling versus bilinear interpolation. While nearest-neighbor upsampling ($2^{nd}$ row) faithfully displays the raw, block-wise assignment at the feature map resolution ($7 \times 7$), it can be visually jarring and difficult to interpret in the context of the high-resolution original image. Bilinear interpolation ($1^{st}$ row) provides a smoother, more interpretable heatmap that better represents the effective receptive field and the continuous nature of the fuzzy membership degrees. We utilize bilinear interpolation in our main paper visualizations to ensure visual clarity without altering the underlying semantic distribution.

**Multi-Head Visualization.** Fig. 5 illustrates the assignment maps from different heads within the last block of stage 3. Similar to multi-head attention, our multi-head clustering mechanism learns distinct structural decompositions of the same input. As shown, different heads exhibit moderate variations in their partitional assignment maps. Furthermore, while fuzzy assignment maps of the same cluster index often capture broadly overlapping regions of interest, they distinguish themselves with distinct focal centers and attention boundaries. This confirms that the multi-head design effectively diversifies the structural representations, preventing redundancy across parallel clustering subspaces.

**Full Cluster Assignment Maps.** Fig. 6 provides a comprehensive visualization of all cluster assignment maps generated by the last block of Stage 3 and Stage 4. We display the fuzzy assignment maps for all clusters (*e.g.*, C1∼C9 in stage 4) alongside the corresponding partitional assignment map. This complete set reveals the full spectrum of learned structural patterns, showcasing how the ensemble covers the entire image content through a diverse set of specialized regions.

## E. Architecture

The architecture of ENFORMER is organized into four hierarchical stages. Each stage consists of a Point Aggregation module, which utilizes a strided convolution to aggregate point features for spatial downsampling, followed by a sequence of EnsembleBlocks. Inside each EnsembleBlock, we integrate our core Deep Ensemble Clustering (DEC) module (see §3.2) with a Feed-Forward Network (FFN). To construct a more robust and efficient backbone, we employ re-parameterization techniques to enhance foundational representational capacity and inference speed. This includes an FFN equipped with re-parameterized depthwise convolutions for enhanced local modeling and the fusion of computational processes during inference to minimize overhead. To further accelerate inference, we utilize linear interpolation in the first two stages to rescale feature maps within the DEC module. This effectively mitigates the computational latency incurred by global similarity calculations on large-scale feature maps. Detailed specifications for ENFORMER variants are presented in Table 15, covering block configurations, feature dimensions, and stage connectivity. Additionally, pseudo-code for our base clustering methods and the DEC module is provided in Algorithms S1 and S2.

*Table 15.* Detailed configurations for ENFORMER. $k$, $s$, and $d$ represent the kernel size, stride, and output dimension in convolution (Conv). $h$ and $hd$ denote the number of heads and head dimension. $cp$ and $cf$ represent the cluster numbers for the "Partitional" and "Fuzzy" methods in the EnsembleBlock. $r$ and $dr$ denote the expansion ratio and dropout ratio in the FFN.

| Stage | Points | ENFORMER-Small | ENFORMER-Base | ENFORMER-Large |
|---|---|---|---|---|
| S1 | 50176 | $\begin{bmatrix} \text{Conv } k3 \ s2 \ d20 \\ \text{Conv } k3 \ s2 \ d40 \end{bmatrix}$ | $\begin{bmatrix} \text{Conv } k3 \ s2 \ d32 \\ \text{Conv } k3 \ s2 \ d64 \end{bmatrix}$ | $\begin{bmatrix} \text{Conv } k3 \ s2 \ d32 \\ \text{Conv } k3 \ s2 \ d64 \end{bmatrix}$ |
| | 3136 | $\begin{bmatrix} \text{DEC } h1 \ hd20 \ cp25 \ cf36 \\ \text{FFN } k3 \ r4 \ dr0 \end{bmatrix} \times 2$ | $\begin{bmatrix} \text{DEC } h1 \ hd32 \ cp25 \ cf36 \\ \text{FFN } k3 \ r4 \ dr0 \end{bmatrix} \times 2$ | $\begin{bmatrix} \text{DEC } h1 \ hd32 \ cp25 \ cf36 \\ \text{FFN } k3 \ r4 \ dr0 \end{bmatrix} \times 2$ |
| S2 | 3136 | $\begin{bmatrix} \text{Conv } k3 \ s2 \ d80 \end{bmatrix}$ | $\begin{bmatrix} \text{Conv } k3 \ s2 \ d128 \end{bmatrix}$ | $\begin{bmatrix} \text{Conv } k3 \ s2 \ d128 \end{bmatrix}$ |
| | 784 | $\begin{bmatrix} \text{DEC } h2 \ hd20 \ cp16 \ cf25 \\ \text{FFN } k3 \ r4 \ dr0 \end{bmatrix} \times 2$ | $\begin{bmatrix} \text{DEC } h2 \ hd32 \ cp16 \ cf25 \\ \text{FFN } k3 \ r4 \ dr0 \end{bmatrix} \times 2$ | $\begin{bmatrix} \text{DEC } h2 \ hd32 \ cp16 \ cf25 \\ \text{FFN } k5 \ r4 \ dr0 \end{bmatrix} \times 3$ |
| S3 | 784 | $\begin{bmatrix} \text{Conv } k3 \ s2 \ d160 \end{bmatrix}$ | $\begin{bmatrix} \text{Conv } k3 \ s2 \ d320 \end{bmatrix}$ | $\begin{bmatrix} \text{Conv } k3 \ s2 \ d320 \end{bmatrix}$ |
| | 196 | $\begin{bmatrix} \text{DEC } h4 \ hd20 \ cp9 \ cf16 \\ \text{FFN } k3 \ r4 \ dr0 \end{bmatrix} \times 10$ | $\begin{bmatrix} \text{DEC } h4 \ hd32 \ cp9 \ cf16 \\ \text{FFN } k5 \ r4 \ dr0 \end{bmatrix} \times 6$ | $\begin{bmatrix} \text{DEC } h5 \ hd32 \ cp9 \ cf16 \\ \text{FFN } k7 \ r4 \ dr0 \end{bmatrix} \times 14$ |
| S4 | 196 | $\begin{bmatrix} \text{Conv } k3 \ s2 \ d320 \end{bmatrix}$ | $\begin{bmatrix} \text{Conv } k3 \ s2 \ d512 \end{bmatrix}$ | $\begin{bmatrix} \text{Conv } k3 \ s2 \ d512 \end{bmatrix}$ |
| | 49 | $\begin{bmatrix} \text{DEC } h8 \ hd20 \ cp4 \ cf9 \\ \text{FFN } k3 \ r4 \ dr0 \end{bmatrix} \times 4$ | $\begin{bmatrix} \text{DEC } h8 \ hd32 \ cp4 \ cf9 \\ \text{FFN } k5 \ r4 \ dr0 \end{bmatrix} \times 2$ | $\begin{bmatrix} \text{DEC } h8 \ hd32 \ cp4 \ cf9 \\ \text{FFN } k7 \ r4 \ dr0 \end{bmatrix} \times 4$ |

## F. Discussion

**Limitation Analysis.** Despite the promising results achieved by ENFORMER, we identify certain limitations in our current implementation. One primary constraint is the fixed combination of base clustering methods (*e.g.*, Partitional + Fuzzy) used during training and inference. While our ablations confirm this static configuration is robust across general benchmarks, it lacks the flexibility to dynamically adapt to the specific content of individual input images. For instance, complex scenes with ambiguous boundaries might benefit more from probabilistic modeling, whereas distinct objects might require hard partitioning. Consequently, a static ensemble may not determine the optimal clustering strategy on a per-sample basis.

**Future Work.** We identify several promising directions for future research:

- **Dynamic Ensemble Routing.** Our additional MoE-style experiment (§D.4) shows that adaptive selection is a promising but not yet superior alternative to the current integration-based design. Future work could explore more advanced routing mechanisms, where a learnable gate dynamically adjusts the contribution of different clustering components according to image content, potentially combining adaptive flexibility with consensus-based robustness.

- **Expanding the Clustering Library.** We plan to incorporate a broader spectrum of algorithms, such as Spectral Clustering (Shi & Malik, 2000). In particular, exploring non-parametric methods that do not require a pre-defined number of clusters (*e.g.*, Affinity Propagation (Frey & Dueck, 2007)) would be valuable for adapting to open-world scenarios with varying object densities.

- **Self-Supervised Learning.** Given the inherent structure-aware nature of clustering, ENFORMER is theoretically well-suited for self-supervised learning paradigms, such as Masked Image Modeling. Investigating how ensemble clustering interacts with masked reconstruction objectives could lead to even stronger, scalable learners.

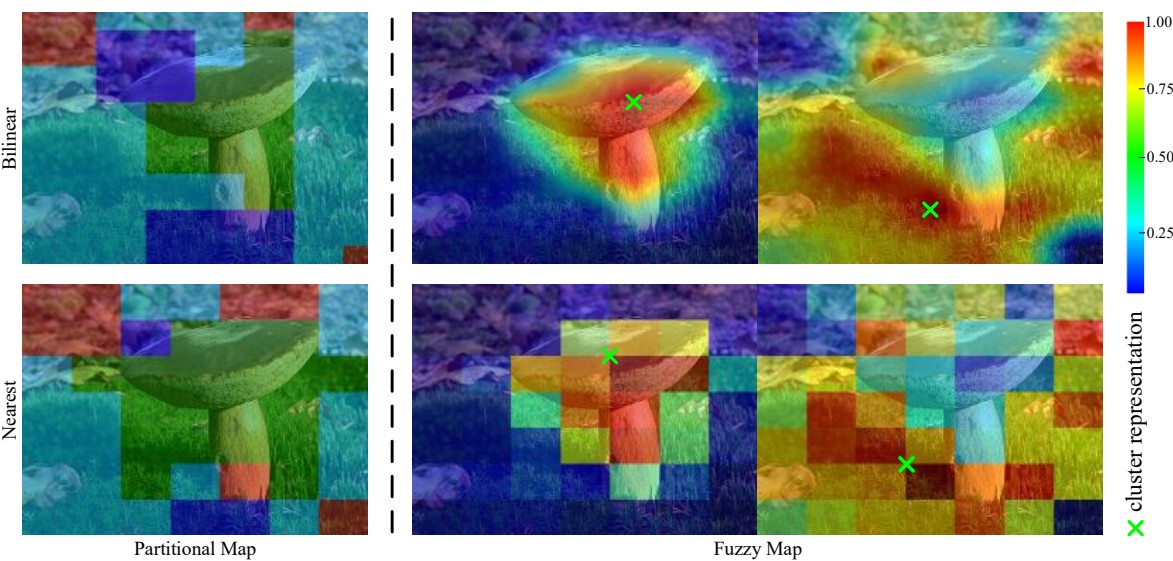

*Figure 4.* Visualization of selected clustering assignment maps with different interpolation methods in the last block of stage 3 (0th head), obtained using ENFORMER-Large.

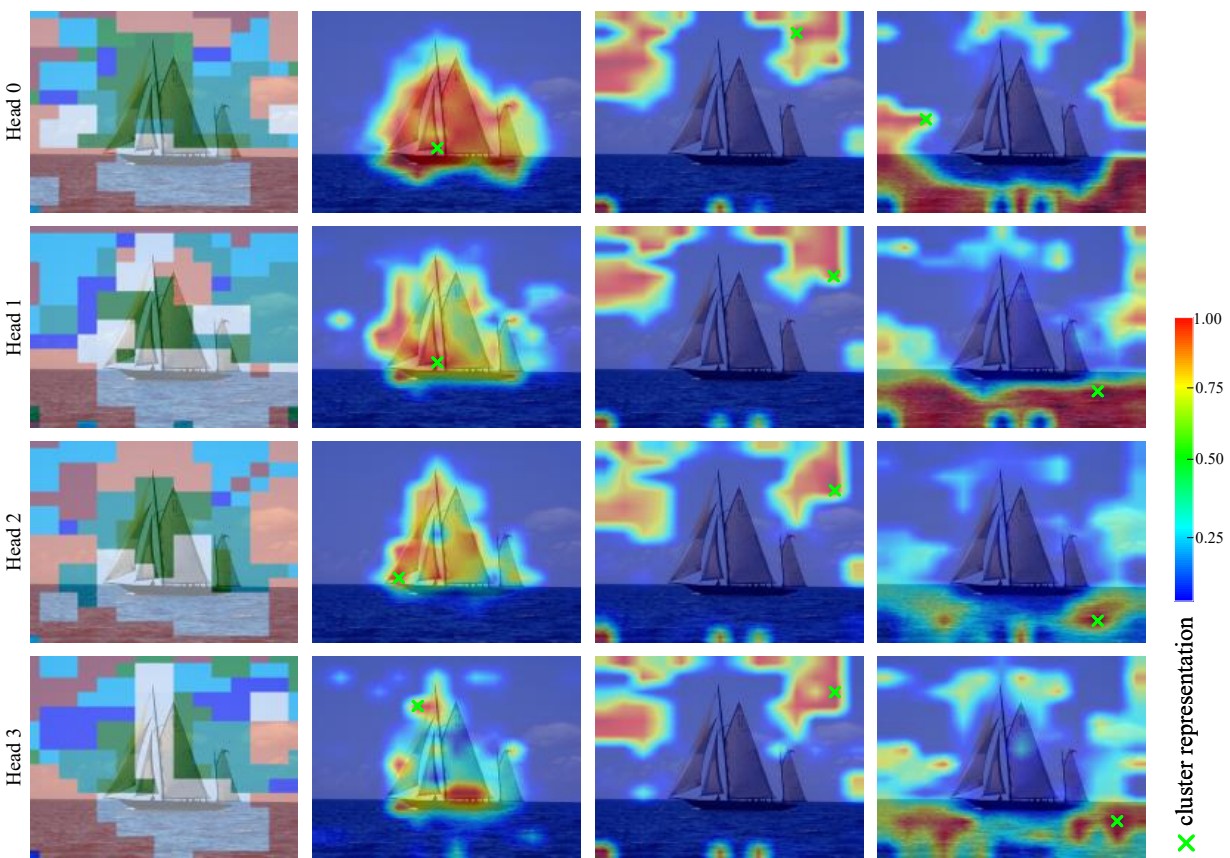

*Figure 5.* Visualization of selected clustering assignment maps from all heads in the last block of stage 3, obtained using ENFORMER-Large.

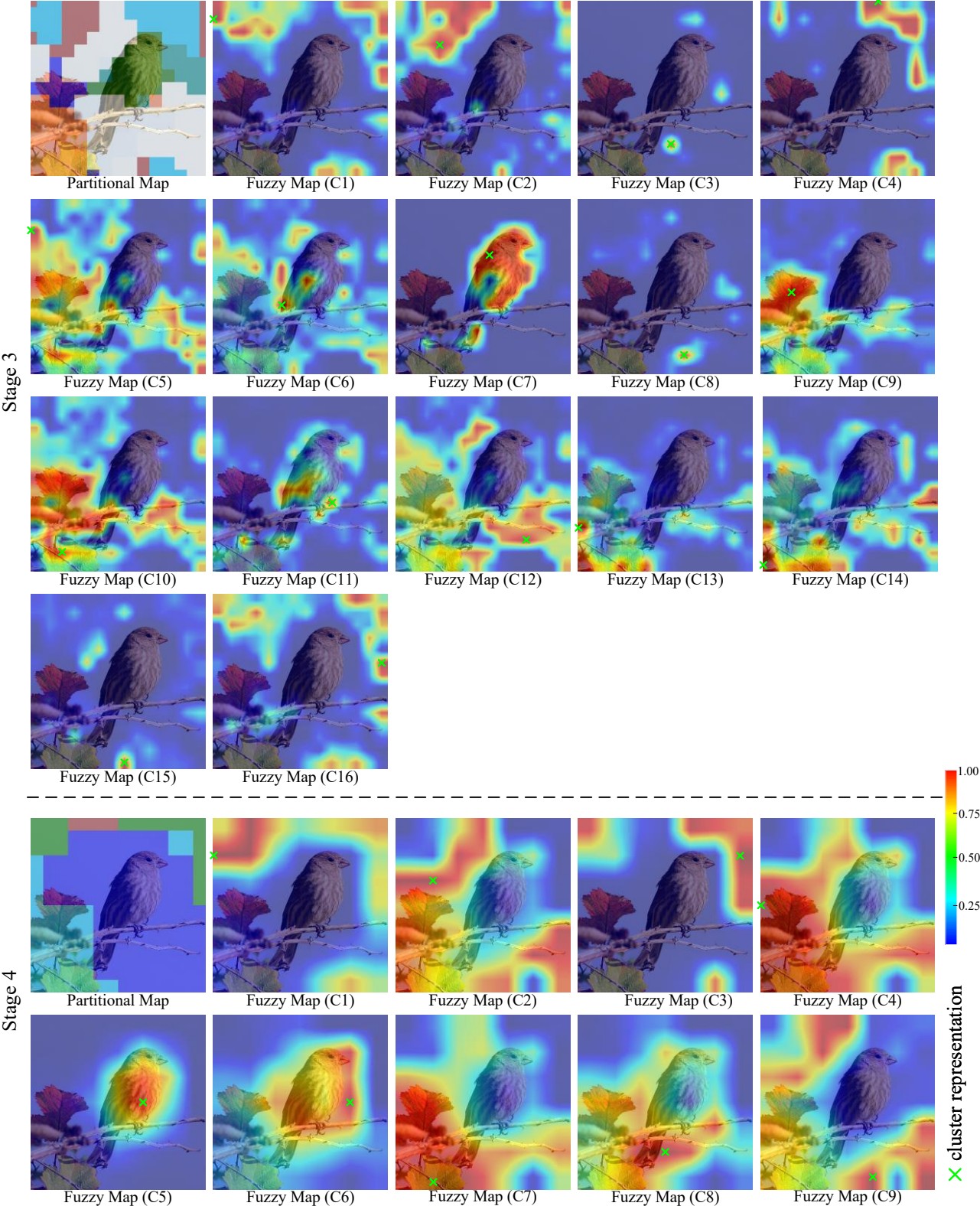

*Figure 6.* Visualization of all clustering assignment maps in the last block of stage 3 and 4, obtained using ENFORMER-Large (3rd head).

**Algorithm S1** Pseudo code of *Base Clustering Methods* in PyTorch-like style.

```
# p, c, v, vc: points [B, N, D], clusters [B, M, D], values [B, N, D], cluster values [B, M, D]
# alpha, beta: learnable parameters for similarity calibration
# log_vars, log_priors: learnable parameters for Probabilistic clustering
# cos_sim: cosine Similarity function; baddbmm(a, b, c): batch matrix-matrix product, computes a + b @ c
# assign: clustering assignment map; agg: aggregated cluster representations

def partitional_clustering(p, c, v, vc): # K-Means style
    sim = alpha * cos_sim(c, p) + beta # [B, M, N]
    mask = gumbel_softmax(sim, hard=True, tau=1.0, dim=1) # [B, M, N]
    assign = sim.sigmoid() * mask # [B, M, N]
    agg = baddbmm(vc, assign, v) / (mask.sum(dim=-1, keepdim=True) + 1.0) # [B, M, D]
    return agg

def fuzzy_clustering(p, c, v, vc): # Fuzzy C-Means style
    sim = alpha * cos_sim(c, p) + beta # [B, M, N]
    assign = softmax(sim, dim=1) # [B, M, N]
    agg = baddbmm(vc, assign, v) / assign.sum(dim=-1, keepdim=True) # [B, M, D]
    return agg

def possibilistic_clustering(p, c, v, vc): # Possibilistic C-Means style
    sim = alpha * cos_sim(c, p) + beta # [B, M, N]
    assign = sim.sigmoid() # Probability independent of clusters # [B, M, N]
    agg = baddbmm(vc, assign, v) / assign.sum(dim=-1, keepdim=True) # [B, M, D]
    return agg

def probabilistic_clustering(p, c, v, vc): # GMM style (E-step only)
    # Compute Mahalanobis distance efficiently
    diff = p.unsqueeze(1) - c.unsqueeze(2) # Broadcasting # [B, M, N, D]
    precision = exp(-log_vars)
    mahalanobis = (diff.pow(2) * precision).sum(dim=-1) # [B, M, N]
    log_likelihood = -0.5 * (mahalanobis + log_vars.sum(dim=-1, keepdim=True)) # [B, M, N]

    # Responsibility: Softmax over likelihoods + priors
    responsibility = softmax(alpha * log_likelihood + beta + log_priors, dim=1) # [B, M, N]
    agg = baddbmm(vc, responsibility, v) / responsibility.sum(dim=-1, keepdim=True) # [B, M, D]
    return agg
```

**Algorithm S2** Pseudo code of *Deep Ensemble Clustering* in PyTorch-like style.

```
# x: input feature map [B, C_in, H, W]
# modules: list of clustering functions (e.g., [Partitional, Fuzzy])
# clusters_pool: list of pooling layers to generate cluster centers from features
# proj_in, proj_out: input and output linear projections
# alpha_ens, beta_ens: learnable parameters for ensemble fusion

def ensemble_clustering(x):
    B, _, H, W = x.shape
    num_mod = len(modules)

    # 1. ensemble generation
    # - p: primary space features r: a set of subspace features for each clustering module
    feats = proj_in(x).view(B, 1 + 2 * num_mod, -1, H, W) # [B, 1+2M, D, H, W]
    p, r = feats[:, 0], feats[:, 1:] # p: [B, D, H, W], r: [B, 2M, D, H, W]

    agg_list = []
    for i in range(num_mod):
        # get points (k) and value (v) features for current module
        k, v = r[:, 2*i], r[:, 2*i+1] # [B, D, H, W]
        # generate cluster centers dynamically via pooling
        k_c = clusters_pool[i](k).flatten(2).permute(0, 2, 1) # [B, M, D]
        v_c = clusters_pool[i](v).flatten(2).permute(0, 2, 1) # [B, M, D]
        # flatten spatial dimensions for points
        k = k.flatten(2).permute(0, 2, 1) # [B, N, D]
        v = v.flatten(2).permute(0, 2, 1) # [B, N, D]
        # execute base clustering module (Algorithm S1)
        agg = modules[i](k, k_c, v, v_c) # [B, M, D]
        agg_list.append(agg)

    # 2. consensus aggregation
    # Concatenate all cluster representatives from all modules
    agg_total = cat(agg_list, dim=1) # [B, M_total, D]

    # weight cluster representatives to reconstruct features
    p_flat = p.flatten(2).permute(0, 2, 1) # [B, N, D]

    # compute similarity between ensemble clusters and primary space point features
    sim = alpha_ens * cos_sim(agg_total, p_flat) + beta_ens # [B, M_total, N]
    assign = softmax(sim, dim=1) # [B, M_total, N]

    # redistribute cluster representations back to points
    out = agg_total.permute(0, 2, 1) @ assign # [B, D, N]
    out = out.view(B, -1, H, W)
    out = proj_out(out) # [B, C_out, H, W]
    return x + out
```

