# OpenReview forum: "Deep Ensemble Clustering for Visual Representation Learning"
_ICML.cc/2026/Conference — ICML 2026 regular_

### Official Review · Reviewer_9ctr · 2026-03-10

**Soundness:** 3
**Presentation:** 3
**Significance:** 2
**Originality:** 2
**Overall Recommendation:** 3
**Confidence:** 4

**Summary:**

This paper proposes ENFORMER, a visual backbone that integrates ensemble clustering directly into feature encoding. Instead of relying on a single clustering mechanism, the method combines multiple differentiable clustering variants during ensemble generation and then fuses them through a differentiable consensus aggregation module. The motivation is that different clustering algorithms capture complementary structural relationships, and their integration can mitigate the bias of any single clustering method.

**Compliance With Llm Reviewing Policy:**

Affirmed.

**Final Justification:**

I thank the authors for the provided response. I keep my original score.

**Key Questions For Authors:**

1. Adaptive selection (such as MOE) is better or integration is better?

2. Could you add a comparison with other types of backbones?

3. Could the authors provide a clearer analysis of the relative contribution of each clustering type, beyond aggregate ablations, to explain why certain heterogeneous combinations are especially effective?

4. How sensitive is ENFORMER to the number of ensemble members and the allocation of feature subspace dimensions across them?

**Limitations:**

Yes. The paper discusses limitations in a reasonably appropriate and constructive way. In particular, the authors acknowledge that the current design uses a fixed combination of clustering methods, rather than dynamically selecting or routing among them based on the input, and they note that simply increasing the number of ensemble members does not necessarily lead to further gains. They also point out that larger ensembles may introduce redundancy or conflicting clustering biases, and that future extensions could explore more adaptive mixture-style designs.

**Strengths And Weaknesses:**

Strengths:

1. The central argument—that single-clustering backbones inherit the bias of one clustering algorithm, while heterogeneous clustering mechanisms may provide complementary structural views—is intuitive and well aligned with the classical ensemble clustering literature.

2. The paper does not merely stack multiple branches, but formulates both ensemble generation and consensus aggregation in a differentiable manner, making the whole architecture trainable end-to-end as a backbone.

3. The model is tested on classification, detection, instance segmentation, and semantic segmentation, which supports the claim that the proposed design is useful as a general-purpose visual backbone rather than a task-specific trick.


4. The paper claims ENFORMER improves over prior clustering-based counterparts while maintaining competitive throughput, which is important for an ensemble-style architecture.

Weaknesses:

1. The main idea is to bring the well-known principle of ensemble clustering into backbone design. While this is a meaningful architectural extension, the paper is stronger as a architecture integration work than as a fundamentally new learning principle.

2. The comparisons are mainly centered on clustering-based backbones. This is reasonable given the paper’s positioning, but it leaves some uncertainty about how competitive the method is relative to stronger contemporary non-clustering backbones under matched scales and training settings.

3. The paper would benefit from deeper analysis of why specific heterogeneous combinations work better.

4. It is not entirely clear how robust the method is to the choice and number of clustering components. Since ensemble methods can be sensitive to redundancy or conflicting inductive biases, a more systematic discussion of when the ensemble helps or saturates would strengthen the paper.

5. Has the author considered whether adaptive selection (such as MOE) is better or integration is better?

---

> ### Author Rebuttal · Authors · 2026-03-31
>
> We thank Reviewer 9ctr for the valuable time and constructive feedback. We provide point-to-point response below.
>
> **Q1: Contribution.**
>
> Thanks for your comment. Our work is motivated by clustering-based vision backbones, an emerging direction, and introduces ensemble clustering to move beyond using only a single clustering algorithm within the backbone. Our contribution is a **new architectural principle** for clustering-based backbones: making ensemble clustering differentiable and end-to-end trainable within the backbone. This allows ensemble generation and consensus aggregation to be integrated directly into feature extraction. We agree that this is not a fundamentally new learning principle, but a meaningful architectural advance for this emerging direction. We will clarify this point in the revision.
>
> **Q2: Comparisons to more backbones.**
>
> Thank you for the helpful suggestion. To broaden the comparison across backbone families, we include one representative model from each of the CNN, SSM, and GNN families, namely ConvNeur [ref1], Mamba® [ref2], and ViG [ref3]. Throughput is measured on a single NVIDIA 4090 GPU with batch size 256. We do not extend this comparison to COCO2017 or ADE20K, as their settings are not fully aligned across methods. Due to limited space, the full table is provided in [Table R4Q2](https://github.com/En-Former/EnFormer/tree/main/rebuttal). EnFormer remains competitive.
>
> **Q3: Why do specific heterogeneous combinations work better.**
>
> Thanks for the question. EnFormer is **insensitive** to ensemble size in a monotonic sense; rather, its behavior depends on whether added components provide distinct structural views. We report the aligned soft-JS between components, which measures their semantic distinctiveness in soft assignment behavior. As shown in [Table R4Q3_1](https://github.com/En-Former/EnFormer/tree/main/rebuttal), both Partitional + Fuzzy and Partitional + Fuzzy + Probabilistic achieve 78.9 Top-1, whereas the all-four setting drops to 78.6 with a much lower soft-JS (0.1948). This suggests that adding more components helps only when they remain complementary, while the gain saturates once they become redundant. Please refer to Reviewer MByD **Q1** for related discussion.
>
> We further add a post-hoc analysis based on the **winner ratio** in the last block, as shown in [Table R4Q3_2](https://github.com/En-Former/EnFormer/tree/main/rebuttal). This metric measures how often each component provides the cluster representative most similar to a primary feature in the final ensemble dictionary, and is used only to understand usage after fusion. In the strongest settings, *Partitional* is the main contributor, while *Fuzzy* and *Probabilistic* still show non-zero usage. This suggests that performance comes from a dominant structural view supported by smaller complementary views. In the *all-four* setting, the extra components are still used, but marginally. Combined with the lower aligned soft-JS, this is consistent with *saturation due to redundancy.*
>
> We will add this in the revision.
>
> **Q4: Sensitivity to the number of ensemble members and the allocation of feature subspace.**
>
> Thanks for your feedback. We analyze this sensitivity from two aspects: clustering components and subspace allocation.
>
> For clustering components, Table 4 shows that EnFormer benefits from moving from a single component to a heterogeneous two-member ensemble, with the best result achieved by Partitional + Fuzzy. Adding a third component remains competitive, but extending to four does not further improve Top-1 accuracy, indicating saturation rather than monotonic gains, i.e., overly large ensembles may introduce redundancy.
>
> For subspace allocation, Eq. 11 aggregates in a shared subspace, so varying per-component dimensions requires projection alignment and is treated here as a stress test rather than the native setting. As shown in [Table R4Q4](https://github.com/En-Former/EnFormer/tree/main/rebuttal), the balanced split (20/20) performs best at 78.9%, while moderate asymmetric splits (16/24 and 24/16) remain competitive at 78.4%, indicating that EnFormer is insensitive to subspace allocation. We will add these results to the revision.
>
> **Q5: Alternative MoE solution.**
>
> Thanks for your suggestion. Per your request, we implemented a minimal MoE-style variant by replacing the original integration-based ensemble module with adaptive selection for fair comparison. The comparison is shown in [Table R4Q5](https://github.com/En-Former/EnFormer/tree/main/rebuttal). The MoE-style variant is 0.7% lower than EnFormer-Base in Top-1 accuracy, suggesting that integration is more effective than adaptive selection under the current design, while the MoE-style result still shows promise. We will add this discussion in the revision.
>
> [ref1] Efficiency Follows Global-Local Decoupling, CVPR26.
>
> [ref2] Mamba®: Vision Mamba ALSO Needs Registers, CVPR25.
>
> [ref3] Vision GNN: An Image is Worth Graph of Nodes, NeurIPS22.

---

> > ### Author Rebuttal · Reviewer_9ctr · 2026-04-04
> >
> > My concerns are partially resolved but inadequate. Clustering-based backbone is developed in recent years. It is not a completely new direction. In my opinion, merely formulating a methodology through an integrated approach is not a particularly impressive contribution for this topic.

---

> > > ### Author Response · Authors · 2026-04-04
> > >
> > > Dear Reviewer 9ctr,
> > >
> > > We sincerely appreciate your time and feedback. To help us improve the work effectively, could you kindly provide more specific details on the remaining unresolved concerns? We are more than willing to perform additional experiments, conduct further analyses, or provide more comprehensive discussions. Thanks.

---

### Official Review · Reviewer_ivY9 · 2026-03-10

**Soundness:** 3
**Presentation:** 4
**Significance:** 3
**Originality:** 4
**Overall Recommendation:** 4
**Confidence:** 3

**Summary:**

This paper proposes ENFORMER, a vision backbone that integrates ensemble clustering into visual representation learning. The method introduces a Deep Ensemble Clustering (DEC) module that combines multiple differentiable clustering mechanisms to capture diverse structural relationships among visual features. The proposed backbone is evaluated on ImageNet-1K, COCO, and ADE20K, demonstrating improved performance over existing clustering-based backbones across classification, detection, and segmentation tasks.

**Compliance With Llm Reviewing Policy:**

Affirmed.

**Final Justification:**

The concerns have been adequately addressed, and my initial assessment remains appropriate. I therefore maintain my recommendation as Weak Accept.

**Key Questions For Authors:**

The authors provided external links (https://github.com/En-Former/EnFormer) in the paper.

Please also refer to the weaknesses section, which summarizes the main issues with this paper.

**Limitations:**

yes

**Strengths And Weaknesses:**

### Strengths

1. The paper proposes a Deep Ensemble Clustering (DEC) module that formulates multiple clustering mechanisms in a differentiable manner, enabling them to be trained jointly within a deep learning framework. The design of ensemble generation and consensus aggregation provides a unified pipeline for combining heterogeneous clustering perspectives.

2. The proposed method is evaluated on three major computer vision benchmarks—ImageNet-1K, COCO, and ADE20K—covering classification, detection, and segmentation tasks. The results demonstrate consistent performance improvements over existing clustering-based backbones across different tasks.

### Weaknesses
1. Throughput is measured on a single V100 GPU, while training is conducted on four A40 GPUs, resulting in an inconsistent hardware setup that may affect the fairness and comparability of the benchmarking results.

2. The paper introduces consensus aggregation involving multiple clustering modules, a cluster dictionary, and query-based reconstruction, but does not provide theoretical complexity or memory analysis, resulting in insufficient methodological analysis.

3. typo: “Partitional Clustering” should also be formatted as a section title.

4. In the fuzzy clustering formulation, the aggregation denominator does not include an additional constant term (as used in Eq. (5) for partitional clustering) to prevent potential numerical instability. The paper does not explain why such a safeguard is unnecessary in this case.

5. The normalization strategy used in aggregation is inconsistent across clustering variants. In **Partitional clustering**, the denominator uses the discretized assignment $X$, whereas in **Fuzzy clustering** it uses the soft assignment matrix $A$. The rationale behind this design choice is unclear and should be clarified.

6. The proposed framework integrates multiple clustering algorithms within the ensemble module. While the paper reports throughput results, it remains unclear how the computational cost scales with the number of clustering components. Could the authors provide a more detailed analysis of the computational and memory overhead introduced by the ensemble clustering design?

---

> ### Author Rebuttal · Authors · 2026-03-31
>
> We thank Reviewer ivY9 for the valuable time and constructive feedback. We provide point-to-point response below.
>
> **Q1: Throughput.**
>
> We thank the reviewer for pointing this out. To ensure consistency, we additionally report throughput on NVIDIA A40 under the same setting (single GPU, batch size 256). Due to limited space, the full table is provided in [Table R3Q1](https://github.com/En-Former/EnFormer/tree/main/rebuttal). We will include these results in the revision.
>
> **Q2: Theoretical complexity analysis on consensus aggregation.**
>
> We thank the reviewer for this valuable comment. We provide the corresponding complexity analysis for consensus aggregation. Let $N$ denote the number of feature points, $d$ the projected subspace dimension, $M^t$ the number of cluster representatives from the $t$-th component, and $M'=\sum_{t=1}^{T}M^t$ the total dictionary size in Eq.11. The query-to-dictionary matching in Eq.12 requires $O(M'Nd)$ time and $O(M'N)$ dominant activation memory. The redistribution in Eq.13 adds another $O(M'Nd)$ term, and the final projection contributes $O(NdD)$, where $D$ is the output channel dimension. Therefore, the overall time complexity is $O(M'Nd+NdD)$, with dominant activation memory $O(M'N)$. We will add this analysis in the revision.
>
> **Q3: Typo.**
>
> Thank you for your careful observation. We will revise the manuscript to ensure consistent formatting of “Partitional Clustering”.
>
> **Q4: Numerical instability in fuzzy clustering.**
>
> Sorry for the confusion. In our implementation, fuzzy aggregation uses a small numerical safeguard by adding ε=1e-7 to the denominator, which was omitted from the manuscript for readability. We will restore this term in the revision for completeness and reproducibility.
>
> **Q5: Normalization strategy used in aggregation.**
>
> Thank you for your careful observation. This difference is **intentional** and is consistent with the definitions of the two clustering formulations. In *Partitional clustering*, the assignment is hard: each sample is assigned to exactly one cluster. Accordingly, the aggregation is computed from the subset of samples selected by the binary assignment matrix (X), and the denominator represents the corresponding **number of assigned samples**. In *Fuzzy clustering*, by contrast, the assignment is soft: each sample may contribute to multiple clusters with different membership weights. The aggregation therefore forms a membership-weighted cluster representative, and the denominator is the **total membership mass** given by the soft assignment matrix (A). We will clarify this more explicitly in the revision.
>
> **Q6: Computational and memory overhead.**
>
> We appreciate this suggestion. Table 4 already partially shows this throughput trend. To further clarify scaling, we also report training and inference memory under a unified setting (NVIDIA A40, batch size 1024, AMP). Due to limited space, the full table is provided in [Table R3Q6](https://github.com/En-Former/EnFormer/tree/main/rebuttal).
>
> As shown in Table R3Q6, the overhead **generally increases with $T$**, reflected by lower throughput and higher training memory. However, it is not ****determined by the number of components alone, and also depends on the **instantiated clustering type**. For example, at $T=2$, Partitional+Probabilistic is slower and more memory-intensive than Partitional+Fuzzy and Partitional+Possibilistic (1377.2 vs. 1476.5/1466.8 img/s; 70.38 vs. 57.51/55.92 GB). A similar trend also appears at $T=3$, where Partitional+Fuzzy+Probabilistic remains slower than Partitional+Fuzzy+Possibilistic (1332.0 vs. 1450.7 img/s). This is because, although these variants have the same asymptotic order, different clustering types introduce **different constant factors** in practice.
>
> Meanwhile, the scaling is **moderated by the shared feature budget**. Increasing $T$ reduces the per-component feature dimension, partially offsetting the cost of adding more components. For example, from Partitional+Probabilistic to Partitional+Fuzzy+Probabilistic, throughput decreases from 1377.2 to 1332.0 img/s, while training memory slightly decreases from 70.38 to 68.85 GB. When $T$ becomes larger, however, the total dictionary size is no longer negligible, so the overhead increases again. The computational trend is consistent with the approximate time complexity $O(\sum_{t=1}^{T}\kappa_t M^tNd+M'Nd)$, where $M'=\sum_{t=1}^{T}M^t$ and $d\approx D'/(1+2T)$, while the training-memory trend is consistent with the approximate dominant memory $O(\sum_{t\in\mathcal{S}}M^tN+\sum_{t\in\mathcal{P}}M^tNd+M'N)$. Hence, the practical overhead is jointly determined by the number of components, the clustering type, and the total dictionary size.
>
> In contrast, inference memory remains **nearly unchanged** across ensemble variants, indicating that the added overhead mainly appears in computation and training-time activations rather than inference-time peak memory. This analysis will be included in the revision.

---

> > ### Author Rebuttal · Reviewer_ivY9 · 2026-04-02
> >
> > His reply answered my question well. The previous rating is well-justified, in my opinion.

---

### Official Review · Reviewer_MByD · 2026-03-11

**Soundness:** 3
**Presentation:** 3
**Significance:** 2
**Originality:** 3
**Overall Recommendation:** 4
**Confidence:** 4

**Summary:**

This work improves the clustering-based vision backbone by including ensemble clustering to mitigate the bias from the individual clustering algorithm. Specifically, some differentiable clustering methods are introduced for ensemble and then a differentiable strategy is developed to fuse the results from each clustering method to refine the visual representation for vision tasks. Experiments on standard benchmark for classification, object detection and segmentation show the effectiveness of the proposed method.

**Compliance With Llm Reviewing Policy:**

Affirmed.

**Final Justification:**

My major concerns have been addressed, and I would like to keep the initial positive rating.

**Key Questions For Authors:**

The motivation that leverages multiple clustering to reduce the bias from individual clustering algorithm is reasonable. However, the empirical study doesn't support the claim where only two clustering methods are helpful for the ensemble. In additional, the efficiency is compared on V100, which is outdated.

**Limitations:**

yes

**Strengths And Weaknesses:**

Strengths
1. The proposed method aims to reduce the bias from the single clustering method by ensemble, which is reasonable.
2. Both individual clustering method and consensus aggregation are differentiable, which is efficient for training.
3. The empirical study shows significant improvement over existing clustering-based vision backbone on benchmark tasks.

Weaknesses
1. While 4 base clustering methods are introduced, only two of them are helpful for the performance as shown in Table 4 while including more base clustering algorithms even hurt the performance, which is not consistent with the motivation of ensemble.
2. According to the comparison in Table 5, the proposed method is sensitive to the number of clusters and different tasks have to tune the appropriate hyper-parameters, which increases the cost of the application for diverse tasks.
3. Throughput is measured on V100, which is the outdated hardware. It is better to report the result on a new hardware, e.g., H100, to provide a more accurate comparison and avoid misleading.

---

> ### Author Rebuttal · Authors · 2026-03-31
>
> We thank Reviewer MByD for the valuable time and constructive feedback. We provide point-to-point response below.
>
> **Q1: Ensemble components.**
>
> Thank you for your comment. We would like to clarify that our claim is NOT that larger ensembles always perform better. Rather, **combining different clustering types is helpful when they provide complementary structural views**. This is consistent with prior ensemble clustering literature: the value comes from reducing single-method bias, while the gain from adding more members is often non-monotonic and can saturate when diversity is limited [ref1-3].
>
> To make this clearer, we analyze component diversity using aligned soft-JS, which measures how distinct the soft assignment behaviors are across clustering components. Higher aligned soft-JS means more distinct views; lower values mean more redundancy. We find that Partitional + Fuzzy and Partitional + Fuzzy + Probabilistic both achieve 78.9 Top-1 with higher aligned soft-JS (0.4485 and 0.3573), while the all-four setting has a lower aligned soft-JS (0.1948) and also lower accuracy (78.6). This suggests that the benefit depends more on diversity than on ensemble size alone.
>
> We will revise the paper to make this claim more precise and include this new analysis.
>
> |Method|Top-1 Acc. (%)|Aligned soft-JS|
> |-|-|-|
> |Partitional+Fuzzy|78.9|0.4485|
> |Partitional+Fuzzy+Probabilistic|78.9|0.3573|
> |all four base clustering methods|78.6|0.1948|
>
> **Q2: Sensitivity to the number of clusters.**
>
> Thank you for your feedback. The main reason for changing the number of clusters across tasks is **the change in input resolution**, rather than sensitivity of the method itself. On ImageNet, Table 5a shows that EnFormer is fairly robust. In Table 5b, COCO and ADE20K are downstream tasks with larger and denser visual inputs, so using more clusters is helpful to preserve finer spatial structure. This is why we use different cluster settings for these tasks. This behavior is also consistent with prior clustering-based backbone [ref4], which uses different cluster numbers across tasks. As a simple practical rule, the number of clusters can be increased together with the feature-map resolution, e.g., by using an adaptive cluster grid proportional to feature-map width/X and height/X. This can reduce manual tuning cost in practice. We will add this discussion to the revision.
>
> **Q3: Throughput.**
>
> Agree! We add efficiency comparisons for ImageNet:
>
> |Method|Param(M)|FLOPs(G)|4090 Throughput(img/s)|Top1-Acc(%)|
> |:-:|:-:|:-:|:-:|:-:|
> |ResNet18|12.0|1.8|6828.8|69.8|
> |ResNet50|26.0|4.1|2144.2|79.8|
> |ConvMixer-512/16|5.4|4.4|858.4|73.8|
> |ConvMixer-1024/12|14.6|13.1|496.8|77.8|
> |ConvMixer-768/32|21.1|5.0|1402.1|80.2|
> |ViT-B/16|86.0|55.5|836.0|77.9|
> |ViT-L/16|307.0|190.7|262.0|76.5|
> |PVT-Tiny|13.2|1.9|3062.6|75.1|
> |PVT-Small|24.5|3.8|1706.2|79.8|
> |Swin-Tiny|29.0|4.5|1499.0|81.3|
> |Swin-Small|50.0|8.7|889.0|83.0|
> |ResMLP-12|15.0|3.0|3226.0|76.6|
> |ResMLP-24|30.0|6.0|1635.7|79.4|
> |ResMLP-36|45.0|8.9|1089.4|79.7|
> |MLP-Mixer-B/16|59.0|12.7|1105.3|76.4|
> |MLP-Mixer-L/16|207.0|44.8|341.8|71.8|
> |gMLP-Ti|6.0|1.4|3728.9|72.3|
> |gMLP-S|20.0|4.5|1637.1|79.6|
> |CoC-Tiny|5.3|1.1|1784.8|71.8|
> |CoC-Small|14.0|2.8|1452.5|77.5|
> |CoC-Medium|27.9|5.9|599.5|81.0|
> |FEC-Small|5.5|1.4|1459.1|72.7|
> |FEC-Base|14.4|3.4|1201.8|78.1|
> |FEC-Large|28.3|6.5|545.4|81.2|
> |**EnFormer-Small(Ours)**|**8.1**|**1.1**|**2735.2**|**78.9**|
> |**EnFormer-Base(Ours)**|**14.8**|**2.5**|**1960.9**|**81.2**|
> |**EnFormer-Large(Ours)**|**29.4**|**4.8**|**1201.9**|**82.6**|
>
> We further report throughput measured on a modern GPU (NVIDIA RTX 4090) under the same evaluation protocol (single GPU with batch size 256). We choose RTX 4090 as it is more widely accessible and representative in current research environments. At the same time, we note that V100 remains a commonly used reference GPU in prior inference benchmarking, which is why it was adopted in our original evaluation. We will include these results in the revision.
>
> [ref1] Cluster Ensembles – A Knowledge Reuse Framework for  Combining Multiple Partitions, JMLR 2002.
>
> [ref2] Weighted cluster ensembles: Methods and analysis, ACM TKDD 2009.
>
> [ref3] DivClust: Controlling Diversity in Deep Clustering, CVPR 2023.
>
> [ref4] Image as set of points, ICLR 2023.

---

> > ### Author Rebuttal · Reviewer_MByD · 2026-04-03
> >
> > My major concerns have been addressed, and I would like to keep the initial positive rating.

---

### Official Review · Reviewer_A4tP · 2026-03-13

**Soundness:** 3
**Presentation:** 3
**Significance:** 3
**Originality:** 2
**Overall Recommendation:** 3
**Confidence:** 3

**Summary:**

This paper proposes ENFORMER, a visual backbone that replaces the usual single clustering mechanism in clustering-based architectures with an ensemble of differentiable clustering modules, followed by a differentiable consensus aggregation step. The method combines several clustering styles, including partitional, fuzzy, possibilistic, and probabilistic variants, and uses their aggregated cluster representations to reconstruct refined features. The paper evaluates ENFORMER on ImageNet-1K classification, COCO detection and instance segmentation, and ADE20K semantic segmentation. Across these benchmarks, the reported results show improvements over prior clustering-based backbones, along with favorable throughput on ImageNet.

**Compliance With Llm Reviewing Policy:**

Affirmed.

**Final Justification:**

My concerns are partially resolved or unresolved, but the remaining concerns are not easily addressed in a short rebuttal. I am going to give final score rating 3.

**Key Questions For Authors:**

\item[KQ1.] You cite recent clustering-based visual learners in the paper, but the experiments mainly compare against CoC and FEC. Why are the other cited clustering-based backbones not included in the empirical evaluation? If you have results, please provide them. This could change my assessment of the paper’s positioning and novelty.
\item[KQ2.] What is the run-to-run variance for the key ablations in Table 4 and for the main ImageNet result in Table 1? If the standard deviation is small relative to the reported margins, that would strengthen the paper considerably.

**Limitations:**

yes

**Strengths And Weaknesses:**

\item[S1.] The paper addresses a notable limitation of prior clustering-based backbones — the reliance on a single clustering inductive bias — and proposes an ensemble clustering solution, which is intuitive both from classical clustering and vision backbone perspectives.
\item[S2.] The empirical coverage extends beyond typical architecture papers, with results on classification, detection, instance segmentation, and semantic segmentation, suggesting that the representations transfer beyond ImageNet.
\item[S3.] The ablation in Table 4 partially supports the core intuition: the Partitional + Fuzzy pair outperforms single-component variants, and Figure 2 clearly distinguishes the ensemble generation and consensus aggregation stages.

\item[W1.] The consensus aggregation (Eqs. 11–13) appears functionally equivalent to cross-attention with keys and values derived from clustered representations. The paper presents this as a novel differentiable consensus mechanism, but it does not formally distinguish it from standard cross-attention, nor does it ablate it against simpler alternatives (e.g., linear projection over concatenated cluster outputs, or simple averaging). Without such comparisons, the source of empirical gains remains unclear.
\item[W2.] The main claim — that ensemble clustering strengthens the representational foundation by mitigating single-method bias — is not fully substantiated. While Table 4 demonstrates that one heterogeneous pair (Partitional + Fuzzy) performs well, results show that T=4 does not outperform T=2, and the best single method is already competitive. The evidence supports only a narrow claim about this specific pair.
\item[W3.] The efficiency claim is only partially substantiated. Throughput is reported for ImageNet classification, but no analysis of latency, memory, or throughput is provided for COCO or ADE20K, where backbone behavior can differ significantly.

---

> ### Author Rebuttal · Authors · 2026-03-31
>
> We thank Reviewer A4tP for the valuable time and constructive feedback. We provide point-to-point response below.
>
> **Q1: Effectiveness of consensus aggregation.**
>
> Thanks for your comment. As you noticed, our contribution lies in introducing a differentiable consensus step for ensemble clustering. Such a design unifies the outputs of multiple base clusterings while implicitly weighting and assessing cluster features. Therefore, the key distinction from cross-attention is semantic: our method **weights cluster-level features and feeds them back to reconstruct point-level features**, rather than weighting point features directly or performing generic token mixing. We will clarify this connection in the revision.
>
> Per your request, we add a fusion baseline that applies a linear projection to the concatenated cluster outputs under the same training setting.
>
> |Aggregation Method|Param (M)|FLOPs (G)|Top1-Acc (%)|
> |-|-|-|-|
> |Linear Projection|7.8|1.1|78.1|
> |Consensus Aggregation|8.1|1.1|78.9|
>
> As seen, this baseline achieves 78.1 (-0.8) Top-1 accuracy on ImageNet-1K. We will include this result in the revision.
>
> **Q2: Ensemble components.**
>
> Thank you for your comment. We would like to clarify that our claim is NOT that larger ensembles always perform better. Rather, **combining different clustering types is helpful when they provide complementary structural views**. This is consistent with prior ensemble clustering literature: the value comes from reducing single-method bias, while the gain from adding more members is often non-monotonic and can saturate when diversity is limited [ref1-3].
>
> To make this clearer, we analyze component diversity using aligned soft-JS, which measures how distinct the soft assignment behaviors are across clustering components. Higher aligned soft-JS means more distinct views; lower values mean more redundancy. We find that Partitional + Fuzzy and Partitional + Fuzzy + Probabilistic both achieve 78.9 Top-1 with higher aligned soft-JS (0.4485 and 0.3573), while the all-four setting has a lower aligned soft-JS (0.1948) and also lower accuracy (78.6). This suggests that the benefit depends more on diversity than on ensemble size alone.
>
> We will revise the paper to make this claim more precise and include this new analysis.
>
> |Method|Top-1 Acc. (%)|Aligned soft-JS|
> |-|-|-|
> |Partitional+Fuzzy|78.9|0.4485|
> |Partitional+Fuzzy+Probabilistic|78.9|0.3573|
> |all four base clustering methods|78.6|0.1948|
>
> **Q3: Efficiency.**
>
> Agree! Due to limited space, the full table is provided in [Table R1](https://github.com/En-Former/EnFormer/tree/main/rebuttal). We report latency, training memory, and inference memory on both datasets under consistent settings (8 GPUs with a total batch size of 16 on NVIDIA A40). On COCO, EnFormer-Small achieves 66.7 ms / 92.53 GB, vs. CoC-Small at 113.3 ms / 181.67 GB and FEC-Small at 112.5 ms / 138.90 GB. On ADE20K, EnFormer-Small achieves 19.38 ms / 35.72 GB, vs. CoC-Small at 40.26 ms / 43.19 GB and FEC-Small at 42.12 ms / 47.61 GB. Comparable trends also hold for the Base and Large variants. Overall, EnFormer shows a clearly improved accuracy-efficiency trade-off across both COCO and ADE20K. We will include these results in the revision.
>
> **Q4: Comparisons to more clustering-based backbones.**
>
> Thank you for your feedback. The submission focused on CoC and FEC because they are the most directly comparable clustering-based backbones in terms of scale and training&evaluation setting. Per your request, we add comparisons with ClusterFormer [ref4] for completeness.
>
> |Method|Param(M)|Top1-Acc(%)|
> |-|-|-|
> |ClusterFormer-Tiny|27.9|81.3|
> |ClusterFormer-Small|48.7|83.4|
> |EnFormer-Small|8.1|78.9|
> |EnFormer-Base|14.8|81.2|
> |EnFormer-Large|29.4|82.6|
>
> As seen, EnFormer remains competitive against ClusterFormer. Under a similar parameter budget (e.g., ClusterFormer-Tiny vs. EnFormer-Large), EnFormer achieves better performance (81.3 vs 82.6). We will include these additional comparisons in the revision.
>
> **Q5: Run-to-run variance.**
>
> Thanks for your suggestion. Due to limited time and computational resources during the rebuttal period, we report error bars for the main results in Table 1 and for selected ablations in Table 4. We will include these results in the revision.
>
> |Method|Top1-Acc(%)|
> |:-:|:-:|
> |EnFormer-Small|78.97±0.06|
> |EnFormer-Base|81.17±0.06|
> |EnFormer-Large|82.60±0.10|
>
> |Ensemble Components|Top1-Acc (%)|
> |-|-|
> | Partitional+Fuzzy|78.97±0.06|
> | Partitional+Probabilistic|78.74±0.07|
> | Partitional+Fuzzy+Possibilistic|78.78±0.09|
>
> [ref1] Cluster Ensembles – A Knowledge Reuse Framework for  Combining Multiple Partitions, JMLR 2002.
>
> [ref2] Weighted cluster ensembles: Methods and analysis, ACM TKDD 2009.
>
> [ref3] DivClust: Controlling Diversity in Deep Clustering, CVPR 2023.
>
> [ref4] Clusterformer: Clustering as a universal visual learner, NeurIPS 2023.

---

> > ### Author Rebuttal · Reviewer_A4tP · 2026-04-01
> >
> > My concerns are partially resolved or unresolved, but the remaining concerns are not easily addressed in a short rebuttal.

---

> > > ### Author Response · Authors · 2026-04-01
> > >
> > > Dear Reviewer A4tP,
> > >
> > > We sincerely appreciate your time and feedback. To help us improve the work effectively, could you kindly provide more specific details on the remaining unresolved concerns? We are more than willing to perform additional experiments, conduct further analyses, or provide more comprehensive discussions. Thanks.

---

### Decision · Program_Chairs · 2026-04-30

**Decision:**

Accept (regular)

**Comment:**

The paper proposes an ensemble of differentiable clustering modules and differentiable consensus aggregation, called ENFORMER, to mitigate the bias from individual clustering algorithm. The paper focuses on algorithm diversity to provide complementary structural views to maintain performance and efficiency.

The reviewers found the proposal of using several clustering methods interesting and the framework of making them differentiable useful.  There are concerns from all about the number of the methods used but the authors address the issue during the rebuttal claiming that the proposal is investigating the impact of such methods instead of the mere addition of them.

After the rebuttal, reviewers MByD and ivY9 recommend a weak accept and have no further concerns.

Reviewer A4tP is against the paper based on claims not being substantiated (ensemble strengthens and efficiency).  The final justification doesn't explain why the concerns were not solved nor which ones.  The authors replied to the raised concerns in their rebuttal.

Reviewer 9ctr maintained after the rebuttal that the technical contribution is limited given that the proposal integrates existing methods instead of proposing a new direction, but recommends a weak accept.

Given the enthusiasm of the reviewers and the broad results, I recommend a weak accept since I agree that the technical contribution is limited but still is interesting.